# FINETUNED LANGUAGE MODELS ARE ZERO-SHOT LEARNERS

**Jason Wei**[*]**, Maarten Bosma**[*]**, Vincent Y. Zhao**[*]**, Kelvin Guu**[*]**, Adams Wei Yu,
Brian Lester, Nan Du, Andrew M. Dai, and Quoc V. Le**

Google Research

## ABSTRACT

This paper explores a simple method for improving the zero-shot learning abilities of language models. We show that *instruction tuning*—finetuning language models on a collection of datasets described via instructions—substantially improves zero-shot performance on unseen tasks.

We take a 137B parameter pretrained language model and instruction tune it on over 60 NLP datasets verbalized via natural language instruction templates. We evaluate this instruction-tuned model, which we call FLAN, on unseen task types. FLAN substantially improves the performance of its unmodified counterpart and surpasses zero-shot 175B GPT-3 on 20 of 25 datasets that we evaluate. FLAN even outperforms few-shot GPT-3 by a large margin on ANLI, RTE, BoolQ, AI2-ARC, OpenbookQA, and StoryCloze. Ablation studies reveal that number of finetuning datasets, model scale, and natural language instructions are key to the success of instruction tuning.

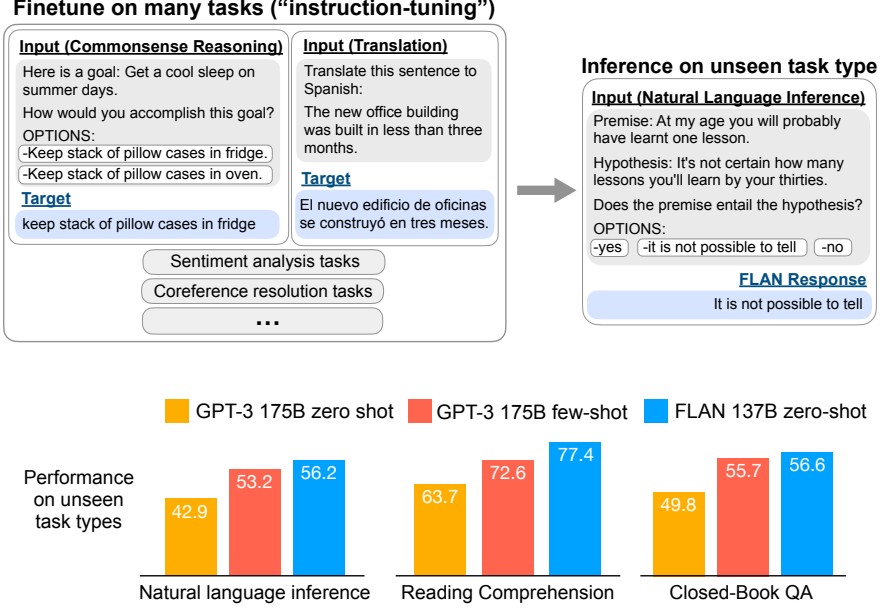

Figure 1: Top: overview of instruction tuning and FLAN. Instruction tuning finetunes a pretrained language model on a mixture of tasks phrased as instructions. At inference time, we evaluate on an unseen task type; for instance, we could evaluate the model on natural language inference (NLI) when no NLI tasks were seen during instruction tuning. Bottom: performance of zero-shot FLAN, compared with zero-shot and few-shot GPT-3, on three unseen task types where instruction tuning improved performance substantially out of ten we evaluate. NLI datasets: ANLI R1–R3, CB, RTE. Reading comprehension datasets: BoolQ, MultiRC, OBQA. Closed-book QA datasets: ARC-easy, ARC-challenge, NQ, TriviaQA.

---

[*]Lead contributors. Author contributions listed at end of paper.

## 1 INTRODUCTION

Language models (LMs) at scale, such as GPT-3 (Brown et al., 2020), have been shown to perform few-shot learning remarkably well. They are less successful at zero-shot learning, however. For example, GPT-3's zero-shot performance is much worse than few-shot performance on tasks such as reading comprehension, question answering, and natural language inference. One potential reason is that, without few-shot exemplars, it is harder for models to perform well on prompts that are not similar to the format of the pretraining data.

In this paper, we explore a simple method to improve the zero-shot performance of large language models, which would expand their reach to a broader audience. We leverage the intuition that NLP tasks can be described via natural language instructions, such as "*Is the sentiment of this movie review positive or negative?*" or "*Translate 'how are you' into Chinese.*" We take a pretrained language model of 137B parameters and perform *instruction tuning*—finetuning the model on a mixture of more than 60 NLP datasets expressed via natural language instructions. We refer to this resulting model as FLAN, for Finetuned Language Net.

To evaluate the zero-shot performance of FLAN on unseen tasks, we group NLP datasets into clusters based on their task types and hold out each cluster for evaluation while instruction tuning FLAN on all other clusters. For example, as shown in Figure 1, to evaluate FLAN's ability to perform natural language inference, we instruction tune the model on a range of other NLP tasks such as commonsense reasoning, translation, and sentiment analysis. As this setup ensures that FLAN has not seen any natural language inference tasks in instruction tuning, we then evaluate its ability to perform zero-shot natural language inference.

Our evaluations show that FLAN substantially improves the zero-shot performance of the base 137B-parameter model. FLAN's zero-shot also outperforms 175B-parameter GPT-3's zero-shot on 20 of 25 datasets that we evaluate, and even outperforms GPT-3's few-shot by a large margin on ANLI, RTE, BoolQ, AI2-ARC, OpenbookQA, and StoryCloze. In ablation studies, we find that increasing the number of task clusters in instruction tuning improves performance on unseen tasks and that the benefits of instruction tuning emerge only with sufficient model scale.

Instruction tuning is a simple method that, as depicted in Figure 2, combines appealing aspects of both the pretrain–finetune and prompting paradigms by using supervision via finetuning to improve language model's responses to inference-time text interactions. Our empirical results demonstrate promising abilities of language models to perform tasks described purely via instructions. Source code for loading the instruction tuning dataset used for FLAN is publicly available at `https://github.com/google-research/flan`.

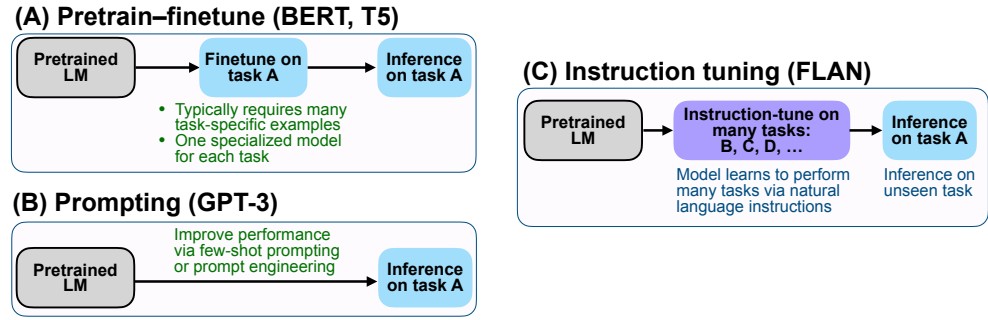

Figure 2: Comparing instruction tuning with pretrain–finetune and prompting.

## 2 FLAN: INSTRUCTION TUNING IMPROVES ZERO-SHOT LEARNING

The motivation of instruction tuning is to improve the ability of language models to respond to NLP instructions. The idea is that by using supervision to teach an LM to perform tasks described via instructions, the LM will learn to follow instructions and do so even for unseen tasks. To evaluate performance on unseen tasks, we group datasets into clusters by task type and hold out each task cluster for evaluation while instruction tuning on all remaining clusters.

## 2.1 TASKS & TEMPLATES

As creating an instruction tuning dataset with many tasks from scratch would be resource-intensive, we transform existing datasets from the research community into an instructional format. We aggregate 62 text datasets that are publicly available on Tensorflow Datasets, including both language understanding and language generation tasks, into a single mixture. Figure 3 shows these datasets—each dataset is categorized into one of twelve task clusters, for which datasets in a given cluster are of the same task type. Descriptions, sizes, and examples of each dataset are shown in Appendix G.

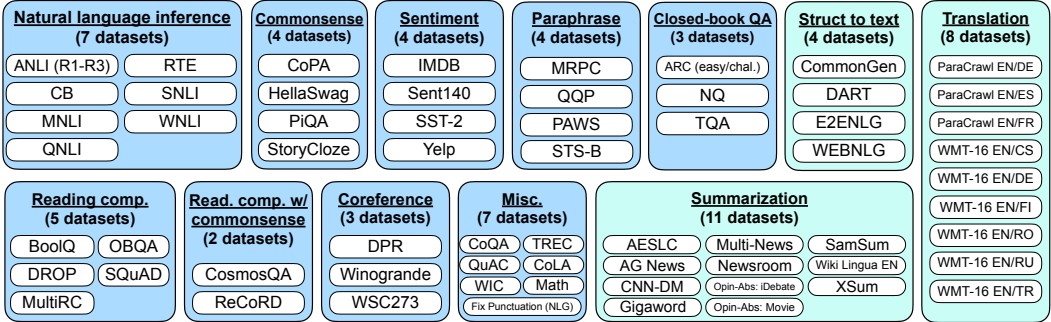

Figure 3: Datasets and task clusters used in this paper (NLU tasks in blue; NLG tasks in teal).

For each dataset, we manually compose ten unique templates that use natural language instructions to describe the task for that dataset. While most of the ten templates describe the original task, to increase diversity, for each dataset we also include up to three templates that "turned the task around," (e.g., for sentiment classification we include templates asking to generate a movie review). We then instruction tune a pretrained language model on the mixture of all datasets, with examples in each dataset formatted via a randomly selected instruction template for that dataset. Figure 4 shows multiple instruction templates for a natural language inference dataset.

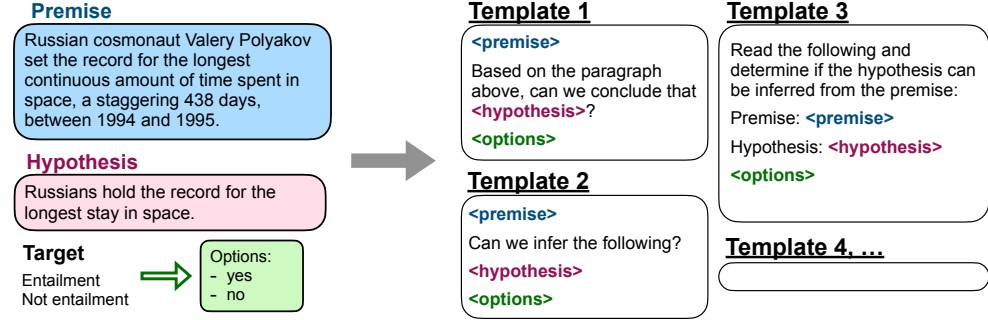

Figure 4: Multiple instruction templates describing a natural language inference task.

## 2.2 EVALUATION SPLITS

We are interested in how FLAN performs on tasks not seen in instruction tuning, and so it is crucial to define what counts as an unseen task. Whereas some prior work defines unseen tasks by disallowing the same dataset to appear in training, we use a more conservative definition that leverages the task clusters from Figure 3. In this work, we only consider dataset $\mathcal{D}$ unseen at evaluation time if no datasets from any task clusters that $\mathcal{D}$ belongs to were seen during instruction tuning. For instance, if $\mathcal{D}$ is an entailment task, then no entailment datasets appeared in instruction tuning, and we instruction-tuned on all other clusters.[1] Hence, to evaluate zero-shot FLAN on $c$ task clusters, we instruction tune $c$ models, where each model holds out a different task cluster for evaluation.

---

[1]When evaluating on the read. comp. with commonsense cluster, both read. comp. and commonsense reasoning were dropped from instruction tuning. Conversely, the read. comp. with commonsense cluster was not used for instruction tuning when evaluating on read. comp. or commonsense reasoning. We also drop the paraphrase cluster from instruction tuning when evaluating on NLI tasks and vice-versa.

## 2.3 CLASSIFICATION WITH OPTIONS

The output space for a given task is either one of several classes (classification) or free text (generation). As FLAN is an instruction-tuned version of a decoder-only language model, it naturally responds in free text, and so no further modifications are needed for generation tasks.

For classification tasks, prior work (Brown et al., 2020) used a *rank classification* approach where, for example, only two outputs ("*yes*" and "*no*") are considered and the higher probability one is taken as the model's prediction. Though this procedure is logically sound, it is imperfect in that the probability mass for answers may have an undesired distribution among ways of saying each answer (e.g., a large number of alternative ways of saying "*yes*" may lower the probability mass assigned to "*yes*"). Therefore, we include an *options* suffix, in which we append the token OPTIONS to the end of a classification task along with a list of the output classes for that task. This makes the model aware of which choices are desired when responding to classification tasks. Example use of options is shown in the NLI and commonsense examples in Figure 1.

## 2.4 TRAINING DETAILS

**Model architecture and pretraining.** In our experiments, we use LaMDA-PT, a dense left-to-right, decoder-only transformer language model of 137B parameters (Thoppilan et al., 2022). This model is pretrained on a collection of web documents (including those with computer code), dialog data, and Wikipedia, tokenized into 2.49T BPE tokens with a 32k vocabulary using the SentencePiece library (Kudo & Richardson, 2018). Around 10% of the pretraining data was non-English. Note that LaMDA-PT only has language model pretraining (c.f. LaMDA, which was finetuned for dialog).

**Instruction tuning procedure.** FLAN is the instruction-tuned version of LaMDA-PT. Our instruction tuning pipeline mixes all datasets and randomly samples from each dataset. To balance the different sizes of datasets, we limit the number of training examples per dataset to 30k and follow the examples-proportional mixing scheme (Raffel et al., 2020) with a mixing rate maximum of 3k.[2] We finetune all models for 30k gradient steps with a batch size of 8,192 tokens using the Adafactor Optimizer (Shazeer & Stern, 2018) with a learning rate of 3e-5. The input and target sequence lengths used in finetuning are 1024 and 256, respectively. We use packing (Raffel et al., 2020) to combine multiple training examples into a single sequence, separating inputs from targets using a special EOS token. This instruction tuning takes around 60 hours on a TPUv3 with 128 cores. For all evaluations, we report results on the final checkpoint trained for 30k steps.

## 3 RESULTS

We evaluate FLAN on natural language inference, reading comprehension, closed-book QA, translation, commonsense reasoning, coreference resolution, and struct-to-text. As described in §2.2, we evaluate on unseen tasks by grouping datasets into task clusters and holding out each cluster for evaluation while instruction tuning on all remaining clusters (i.e., each evaluation task cluster uses a different checkpoint). For each dataset, we evaluate the mean of performance on all templates, which proxies the expected performance given a typical natural language instruction. As a dev set is sometimes available for manual prompt engineering (Brown et al., 2020), for each dataset we also obtain the test set performance using the template with the best dev set performance.

For comparison, we report zero and few-shot results for LaMDA-PT using the same prompts as GPT-3 (as LaMDA-PT is not suitable for natural instructions without instruction tuning). This baseline provides the most direct ablation of how much instruction tuning helps. Instruction tuning significantly improves LaMDA-PT on most datasets.

We also show the zero-shot performances of GPT-3 175B (Brown et al., 2020) and GLaM 64B/64E (Du et al., 2021), as reported in their respective papers. With the best dev template, zero-shot FLAN outperforms zero-shot GPT-3 on 20 of 25 datasets and even surpasses GPT-3's few-shot performance on 10 datasets. With the best dev-template, zero-shot FLAN outperforms zero-shot GLaM on 13 of 19 available datasets and one-shot GLaM on 11 of 19 datasets.

---

[2]In this mixing scheme, a mixing rate maximum of 3,000 means that a dataset does not receive additional sampling weight for examples in excess of 3,000.

Overall, we observe that instruction tuning is very effective on tasks naturally verbalized as instructions (e.g., NLI, QA, translation, struct-to-text) and is less effective on tasks directly formulated as language modeling, where instructions would be largely redundant (e.g., commonsense reasoning and coreference resolution tasks that are formatted as finishing an incomplete sentence or paragraph). Results on natural language inference, reading comprehension, closed-book QA, and translation are summarized in Figure 5 and described below.

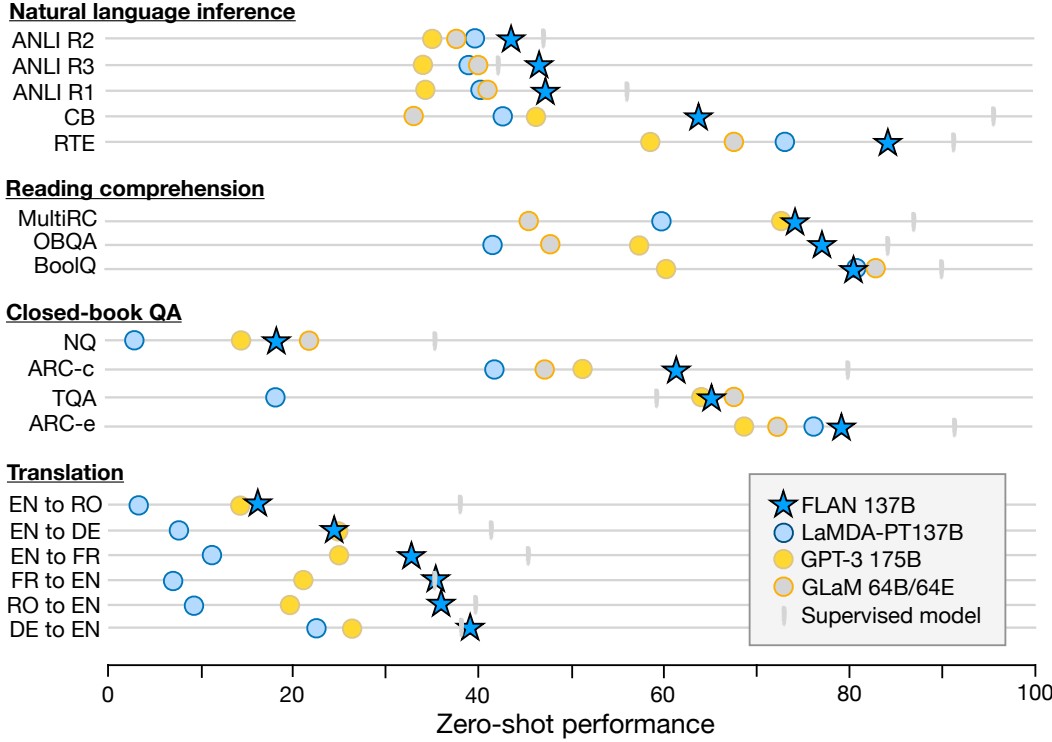

Figure 5: Zero-shot performance of FLAN compared to LaMDA-PT 137B, GPT-3 175B, and GLaM 64B/64E on natural language inference, reading comprehension, closed-book QA, and translation. Performance of FLAN is the mean of up to 10 instructional templates per task. Supervised models were either T5, BERT, or translation models (specified in Table 2 and Table 1 in the Appendix).

**Natural language inference (NLI).** On five NLI datasets, where a model must determine whether a hypothesis is true given some premise, FLAN outperforms all baselines by a large margin. As noted by Brown et al. (2020), perhaps one reason why GPT-3 struggles with NLI is that NLI examples are unlikely to have appeared naturally in an unsupervised training set and are thus awkwardly phrased as a continuation of a sentence. For FLAN, we phrase NLI as the more natural question "`Does <premise> mean that <hypothesis>?`", achieving much higher performance.

**Reading comprehension.** On reading comprehension, where models are asked to answer a question about a provided passage, FLAN outperforms baselines for MultiRC (Khashabi et al., 2018) and OBQA (Mihaylov et al., 2018). On BoolQ (Clark et al., 2019a), FLAN outperforms GPT-3 by a large margin, though LaMDA-PT already achieves high performance on BoolQ.

**Closed-book QA.** For closed-book QA, which asks models to answer questions about the world without access to specific information containing the answer, FLAN outperforms GPT-3 on all four datasets. Compared to GLaM, FLAN has better performance on ARC-e and ARC-c (Clark et al., 2018), and slightly lower performance on NQ (Lee et al., 2019; Kwiatkowski et al., 2019) and TQA (Joshi et al., 2017).

**Translation.** Similar to GPT-3, the training data for LaMDA-PT is around 90% English and includes some text in other languages that was not specifically used to train the model to perform machine translation. We also evaluate FLAN's performance on machine translation for the three datasets evaluated in the GPT-3 paper: French–English from WMT'14 (Bojar et al., 2014), and German–

English and Romanian–English from WMT'16 (Bojar et al., 2016). Compared with GPT-3, FLAN outperforms zero-shot GPT-3 for all six evaluations, though it underperforms few-shot GPT-3 in most cases. Similar to GPT-3, FLAN shows strong results for translating into English and compares favorably against supervised translation baselines. Translating from English into other languages, however, was relatively weaker, as might be expected given that FLAN uses an English sentencepiece tokenizer and that the majority of pretraining data is English.

**Additional tasks.** Although we see strong results for the above task clusters, one limitation with instruction tuning is that it does not improve performance for many language modeling tasks (e.g., commonsense reasoning or coreference resolution tasks formulated as sentence completions). For seven commonsense reasoning and coreference resolution tasks (see Table 2 in the Appendix), FLAN only outperforms LaMDA-PT on three of the seven tasks. This negative result indicates that when the downstream task is the same as the original language modeling pre-training objective (i.e., in cases where instructions are largely redundant), instruction tuning is not useful. Finally, we report results for sentiment analysis, paraphrase detection, and struct-to-text, as well as additional datasets for which GPT-3 results are not available, in Table 2 and Table 1 in the Appendix. Generally, zero-shot FLAN outperforms zero-shot LaMDA-PT and is comparable with or better than few-shot LaMDA-PT.

# 4 ABLATION STUDIES & FURTHER ANALYSIS

## 4.1 NUMBER OF INSTRUCTION TUNING CLUSTERS

As the core question of our paper asks how instruction tuning improves a model's zero-shot performance on unseen tasks, in this first ablation we examine how performance is affected by the number of clusters and tasks used in instruction tuning. For this setup, we hold out NLI, closed-book QA, and commonsense reasoning as evaluation clusters, and use the seven remaining clusters for instruction tuning.[3] We show results for one to seven instruction tuning clusters, where clusters are added in decreasing order of number of tasks per cluster.

Figure 6 shows these results. As expected, we observe that average performance across the three held-out clusters improves as we add additional clusters and tasks to instruction tuning (with the exception of the sentiment analysis cluster), confirming the benefits of our proposed instruction tuning approach on zero-shot performance on novel tasks. It is further interesting to see that, for the seven clusters we test, the performance does not appear to saturate, implying that performance may further improve with even more clusters added to instruction tuning. Of note, this ablation does not allow us to draw conclusions about which instruction tuning cluster contributes the most to each evaluation cluster, although we see minimal added value from the sentiment analysis cluster.

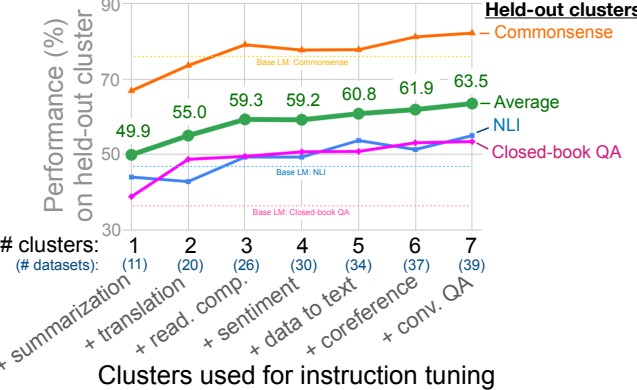

Figure 6: Adding additional task clusters to instruction tuning improves zero-shot performance on held-out task clusters. The evaluation tasks are the following. Commonsense: CoPA, HellaSwag, PiQA, and StoryCloze. NLI: ANLI R1–R3, QNLI, RTE, SNLI, and WNLI. Closed-book QA: ARC easy, ARC challenge, Natural Questions, and TriviaQA.

---

[3]We do not use the paraphrase or reading comprehension with commonsense clusters for instruction tuning in this ablation because they are too similar to NLI and commmonsense reasoning, respectively.

## 4.2 SCALING LAWS

As Brown et al. (2020) shows that zero and few-shot capabilities of language models substantially improve for larger models, we next explore how the benefits of instruction tuning are affected by model scale. Using the same cluster split as in the previous ablation study, we evaluate the effect of instruction tuning on models of size 422M, 2B, 8B, 68B, and 137B parameters.

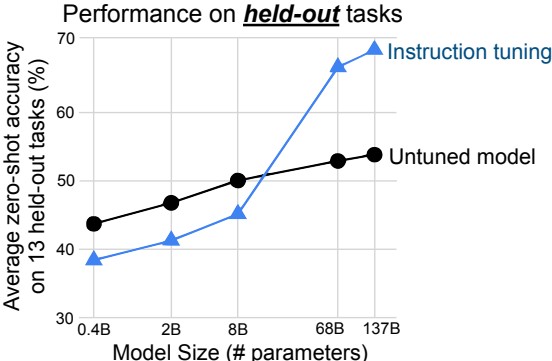

Figure 7 shows these results. We see that for the two models on the order of 100B parameters, instruction tuning substantially improves performance on held-out tasks, as is expected given the prior results in our paper. The behavior on held-out tasks for the 8B and smaller models, however, is thought-provoking—instruction tuning actually hurts performance on held-out tasks. One potential explanation for this result could be that for small-scale models, learning the ∼40 tasks used during instruction tuning fills the entire model capacity, causing these models to perform worse on new tasks. Under this potential explanation, for the larger scale models, instruction tuning fills up some model capacity but also teaches these models how to follow instructions, allowing them to generalize to new tasks with the remaining capacity.

Figure 7: Whereas instruction tuning helps large models generalize to new tasks, for small models it actually hurts generalization to unseen tasks, potentially because all model capacity is used to learn the mixture of instruction tuning tasks.

## 4.3 ROLE OF INSTRUCTIONS

In a final ablation study, we explore the role of instructions during finetuning, as one possibility is that performance gains come entirely from multi-task fine-tuning and the model could perform just as well without instructions. We hence consider two finetuning setups without instructions. In a *no template* setup, only inputs and outputs were given to the model (e.g., for translation the input would be "*The dog runs.*" and the output would be "*Le chien court.*"). In a *dataset name* setup, each input is prepended with the name of the task and dataset (e.g., for translation to French, the input would be "*[Translation: WMT'14 to French] The dog runs.*").

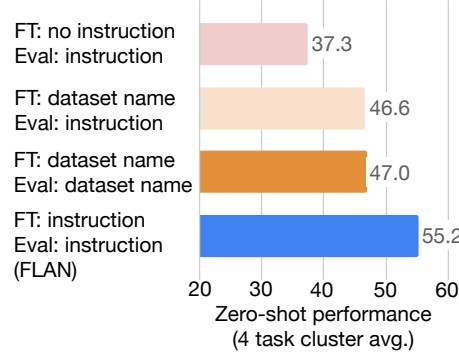

Figure 8: Ablation study result using models with instructions removed from finetuning (FT).

We compare these two ablations to FLAN's finetuning procedure, which used natural instructions (e.g., "*Please translate this sentence to French: 'The dog runs.'*"). We perform evaluations for four held-out clusters from Figure 5. For the no template setup, we used the FLAN instructions during zero-shot inference (because if we used no template, the model would not know what task to perform). For models finetuned on dataset name only, we report zero-shot performance for FLAN instructions as well as using the dataset name. Figure 8 shows the results—both ablation configurations performed substantially worse than FLAN, indicating that training with instructions is crucial for zero-shot performance on unseen tasks.

## 4.4 INSTRUCTIONS WITH FEW-SHOT EXEMPLARS

So far, we have focused on instruction tuning in the zero-shot setting. Here, we study how instruction tuning can be used when few-shot exemplars are available at inference time. The format for the few-shot setting builds on the zero-shot format. For some input $x$ and output $y$, let instruct($x$) denote the zero-shot instructions. Then, given $k$ few-shot exemplars $(x_i, y_i)_{i=1}^{k}$ and a new input $x$, the instruction format for the few-shot setting is "instruct($x_1$) $\oplus$ $y_1$ $\oplus$ instruct($x_2$) $\oplus$ $y_2$ $\oplus$ ... $\oplus$

instruct$(x_k) \oplus y_k \oplus$ instruct$(x)$", where $\oplus$ denotes string concatenation with a delimiter token inserted in between. At both training and inference time, exemplars are randomly drawn from the training set, and the number of exemplars is capped at 16 and such that the total sequence length is less than 960 tokens. Our experiment uses the same task splits and evaluation procedure as §3, such that few-shot exemplars for an unseen task are only used at inference time.

As shown in Figure 9, few-shot exemplars improve the performance on all task clusters, compared with zero-shot FLAN. Exemplars are especially effective for tasks with large/complex output spaces, such as struct to text, translation, and closed-book QA, potentially because exemplars help the model better understand the output format. In addition, for all task clusters, standard deviation among templates is lower for few-shot FLAN, indicating reduced sensitivity to prompt engineering.

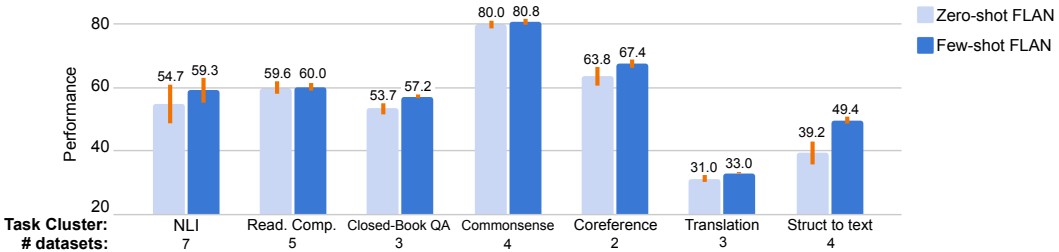

Figure 9: Adding few-shot exemplars to FLAN is a complementary method for improving the performance of instruction-tuned models. The orange bars indicate standard deviation among templates, averaged at the dataset level for each task cluster.

### 4.5 INSTRUCTION TUNING FACILITATES PROMPT TUNING

As we've seen that instruction tuning improves the ability of a model to respond to instructions, it follows that, if FLAN is indeed more amenable to performing NLP tasks, then it should also achieve better performance when performing inference using soft prompts, represented by prepended continuous variables optimized via prompt tuning (Li & Liang, 2021; Lester et al., 2021). As further analysis, we train continuous prompts for each of the SuperGLUE (Wang et al., 2019a) tasks in accordance with the cluster splits from §2.2 such that when prompt-tuning on task $\mathcal{T}$, no tasks in the same cluster as $\mathcal{T}$ were seen during instruction tuning. Our prompt tuning setup follows the procedure of Lester et al. (2021) except that we use a prompt length of 10, weight decay of 1e-4, and did not use dropout on the attention scores; we found in preliminary experiments that these changes improved the performance of LaMDA-PT.

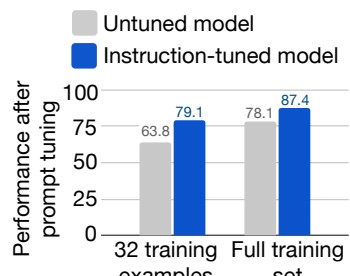

Figure 10: Instruction-tuned models respond better to continuous inputs from prompt tuning. When prompt tuning on a given dataset, no tasks from the same cluster as that dataset were seen during instruction tuning. Performance shown is the average on the SuperGLUE dev set.

Figure 10 shows the results of these prompt tuning experiments for both using a fully-supervised training set and in a low-resource setting with only 32 training examples. We see that in all scenarios, prompt tuning works better with FLAN than LaMDA-PT. In many cases, especially for the low-resource setting, prompt tuning on FLAN even achieves more than 10% improvement over prompt tuning on the LaMDA-PT. This result exemplifies in another way how instruction tuning can result in a checkpoint that is more desirable for performing NLP tasks.

## 5 RELATED WORK

Our work relates to several broad research areas including zero-shot learning, prompting, multi-task learning, and language models for NLP applications (Radford et al., 2019; Raffel et al., 2020; Brown et al., 2020; Efrat & Levy, 2020; Aghajanyan et al., 2021; Li & Liang, 2021, *inter alia*). We describe prior work for these broad areas in an extended related work section (Appendix D), and here we describe two subareas narrower in scope that perhaps relate most closely to our work.

The way we ask a model to respond to instructions is similar to QA-based task formulation (Kumar et al., 2016; McCann et al., 2018), which aims to unify NLP tasks by casting them as QA over a context. Though these methods are very similar to ours, they mostly focus on multi-task learning instead of zero-shot learning, and—as noted by Liu et al. (2021)—they are generally not motivated by using existing knowledge in pretrained LMs. Moreover, our work supercedes recent work such as Chai et al. (2020) and Zhong et al. (2021) in terms of both model scale and scope of tasks.

The success of language models has led to nascent research on the ability of models to follow instructions. Most recently, Mishra et al. (2021) finetune 140M parameter BART on instructions with few-shot exemplars, and evaluate its few-shot abilities on unseen tasks—this is similar to our few-shot instruction tuning result from §4.4. This promising result (as well as one from Ye et al. (2021), which does not emphasize instructions as much) suggests that finetuning on a collection of tasks improves few-shot performance on unseen tasks, even at a smaller model scale. Sanh et al. (2021) finetune T5 in a setup similar to ours, finding that zero-shot learning can be improved in a model of 11B parameters. At a model scale similar to ours, OpenAI's InstructGPT models are trained via both finetuning and reinforcement learning to produce outputs that are more preferred by human raters (Ouyang et al., 2022).

## 6 DISCUSSION

Our paper has explored a simple question in zero-shot prompting: does finetuning a model on a collection of tasks phrased as instructions improve its performance on unseen tasks? We operationalize this question via instruction tuning, a simple method that combines appealing aspects of both the pretrain–finetune and prompting paradigms. Our instruction-tuned model, FLAN, improves performance against an untuned model and surpasses zero-shot GPT-3 on the majority of tasks that we evaluate on. Ablation studies reveal that performance on unseen tasks improves with the number of instruction tuning task clusters, and, interestingly, that performance improvements from instruction tuning emerge only with sufficient model scale. Moreover, instruction tuning can be combined with other prompting methods such as few-shot prompting and prompt tuning.

The diverse capabilities of language models at scale have drawn attention to the tradeoffs between specialist models (one model per task) and generalist models (one model for many tasks; Arivazhagan et al., 2019; Pratap et al., 2020), for which our study has potential implications. Although one might expect labeled data to have the most natural role in improving specialist models, instruction tuning demonstrates how labeled data can be used to help large language models perform many, unseen tasks. In other words, the positive effect of instruction tuning on cross-task generalization shows that task-specific training is complementary to general language modeling and motivates further research on generalist models.

As for limitations of our study, there is a degree of subjectivity in assigning tasks to clusters (though we try to use accepted categorizations in the literature), and we only explore the use of relatively short instructions of typically a single sentence (c.f. detailed instructions given to crowd-workers). A limitation for our evaluation is that individual examples might have appeared in the models' pretraining data, which includes web documents, though in post-hoc analysis (Appendix C) we do not find any evidence that data overlap substantially impacted the results. Finally, the scale of FLAN 137B makes it costly to serve. Future work on instruction tuning could include gathering/generating even more task clusters for finetuning, cross-lingual experiments, using FLAN to generate data for training downstream classifiers, and using finetuning to improve model behavior with respect to bias and fairness (Solaiman & Dennison, 2021).

## 7 CONCLUSIONS

This paper has explored a simple method for improving the ability of language models at scale to perform zero-shot tasks based purely on instructions. Our instruction-tuned model, FLAN, compares favorably against GPT-3 and signals the potential ability for language models at scale to follow instructions. We hope that our paper will spur further research on instructions-based NLP, zero-shot learning, and using labeled data to improve large language models.

ETHICAL CONSIDERATIONS

This work uses language models, for which the risks and potential harms are discussed in Bender & Koller (2020), Brown et al. (2020), Bender et al. (2021), Patterson et al., (2021), and others. As our contribution in this paper is not a pretrained language model itself but rather an empirical study of how instruction tuning affects the zero-shot performance of a language model on unseen tasks, we additionally highlight two relevant ethical considerations. First, labeled datasets such as those we use for finetuning can contain undesirable biases, and these biases can be propagated into zero-shot applications of the model on downstream tasks. And second, instruction-tuned models can potentially require less data and expertise to use; such lower barriers to access could increase both the benefits and associated risks of such models.

ENVIRONMENTAL CONSIDERATIONS

We use the same pretrained language models as Austin et al. (2021). The energy cost and carbon footprint for the pretrained models were 451 MWh and 26 tCO2e, respectively. The additional instruction tuning gradient-steps for finetuning FLAN is less than 2% of the number of pretraining steps, and so the estimated additional energy cost is comparatively smaller.

AUTHOR CONTRIBUTIONS

Maarten Bosma conceived the original idea and implemented the first version of FLAN. Vincent Zhao prototyped the training and evaluation pipelines, as well as rank classification. Kelvin Guu proposed and implemented the idea of task clusters and evaluation using inter-cluster splits. Jason Wei, Maarten Bosma, Vincent Zhao, and Adams Wei Yu implemented the NLP tasks. Jason Wei, Vincent Zhao, and Adams Wei Yu conducted and managed most of the experiments. Jason Wei designed and ran the ablation studies. Jason Wei, Maarten Bosma, and Quoc V. Le wrote most of the paper. Jason Wei, Maarten Bosma, and Nan Du obtained the zero and few-shot baselines. Vincent Zhao and Kelvin Guu designed, implemented, and conducted the few-shot FLAN experiments. Maarten Bosma and Jason Wei ran the data contamination analysis. Brian Lester ran the prompt tuning experiments. Quoc V. Le and Andrew M. Dai advised, provided high-level guidance, and helped edit the paper.

ACKNOWLEDGEMENTS

We thank Ed Chi, Slav Petrov, Dan Garrette, Ruibo Liu, and Clara Meister for providing feedback on our manuscript. We thank Adam Roberts, Liam Fedus, Hyung Won Chung, and Noam Shazeer for helping debug some of our models. We thank Ellie Pavlick for feedback on the study design during the middle stages of the project. We thank Daniel De Freitas Adiwardana for helping initiate the project, large language model advising, and giving us access to some computational resources. Finally, we thank the team involved in pretraining LaMDA-PT: Daniel De Freitas Adiwardana, Noam Shazeer, Yanping Huang, Dmitry Lepikhin, Dehao Chen, Yuanzhong Xu and Zhifeng Chen.

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

## A  ADDITIONAL RESULTS

This section shows the full results for all datasets we evaluate. Results for translation and struct to text are shown in Table 1, and the results for eight NLU task clusters are shown in Table 2.

We show FLAN's performance using the best of up to ten instruction templates as well as the template with the best performance on the dev set. For LaMDA-PT, we use the templates from Brown et al. (2020), which were optimized for GPT-3, without performing any prompt engineering to optimize them on our model. For simplicity, we use greedy search for all generative tasks (compared with beam search used in Brown et al. (2020)). Unlike GPT-3, which chooses the number of few-shot exemplars $k$ via best dev set performance, for few-shot LaMDA-PT we choose the highest $k$ that fits in the context length of 1024 tokens, from $k \in \{1, 3, 5, 10\}$.

For DROP (Dua et al., 2019) and SQuADv2 (Rajpurkar et al., 2018), based on email correspondence with Brown et al. (2020), their definition of zero-shot differs from ours in that they actually use exemplars, but only from the same passage as the inference question (each passage has more than one question). Hence, GPT-3 zero-shot results are not directly comparable with ours for DROP and SQuADv2. We mark these results using the [†] symbol. Moreover, it is unclear how to parse the end of an answer for these two datasets, and so we use curly bracket delimiters { and }, where we expect } to indicate the end of the answer.

For struct to text, reported T5/mT5 results are from the GEM benchmark paper (Gehrmann et al., 2021), though we do not report their results for DART (through correspondence with authors, we confirmed that their results for DART were incorrect). Though we use a summarization task cluster during instruction tuning, we leave evaluation of summarization for future work, as the mean input of most summarization datasets exceeds FLAN's input length of 1024 tokens.

| | | | LaMDA-PT | | GPT-3 175B | | FLAN 137B | | | | | |
| | | | | | | | zero-shot | | few-shot | | | |
| | Metric | Supervised Model | zero-shot | few-shot [k] | zero-shot | few-shot [k] | average template | best dev template | average template | best dev template | [k] | #t |
|---|---|---|---|---|---|---|---|---|---|---|---|---|
| **TRANSLATION** | | | | | | | | | | | | |
| WMT '14 En→Fr | BLEU | $35.0^d$ | 11.2 | 31.5 [5] | 25.2 | 32.6 [64] | $32.9_{\pm 1.1}$ | 33.9 | $33.9_{\pm 0.2}$ | 33.8 | [9] | 5 |
| WMT '14 Fr→En | BLEU | $45.6^c$ | 7.2 | 34.7 [5] | 21.2 | 39.2 [64] | $35.5_{\pm 1.3}$ | 35.9 | $38.0_{\pm 0.1}$ | 37.9 | [9] | 3 |
| WMT '16 En→De | BLEU | $38.6^f$ | 7.7 | 26.7 [5] | 24.6 | 29.7 [64] | $25.4_{\pm 1.8}$ | 27.0 | $26.8_{\pm 0.4}$ | 26.1 | [11] | 5 |
| WMT '16 De→En | BLEU | $41.2^e$ | 20.8 | 36.8 [5] | 27.2 | 40.6 [64] | $38.9_{\pm 0.3}$ | 38.9 | $40.6_{\pm 0.1}$ | 40.7 | [11] | 3 |
| WMT '16 En→Ro | BLEU | $39.9^g$ | 3.5 | 22.9 [5] | 14.1 | 21.0 [64] | $16.7_{\pm 1.6}$ | 18.9 | $20.5_{\pm 0.1}$ | 20.5 | [9] | 5 |
| WMT '16 Ro→En | BLEU | $38.5^g$ | 9.7 | 37.5 [5] | 19.9 | 39.5 [64] | $36.8_{\pm 0.5}$ | 37.3 | $38.2_{\pm 0.1}$ | 38.1 | [9] | 3 |
| **STRUCT TO TEXT** | | | | | | | | | | | | |
| CommonGen | Rouge-1 | $64.0^a$ | 3.9 | 56.7 [3] | – | – | $54.6_{\pm 2.3}$ | 56.3 | $56.6_{\pm 0.3}$ | 56.4 | [16] | 6 |
| | Rouge-2 | $29.4^a$ | 1.5 | 29.6 [3] | – | – | $28.8_{\pm 2.4}$ | 27.6 | $30.9_{\pm 0.7}$ | 29.9 | [16] | 6 |
| | Rouge-L | $54.5^a$ | 3.2 | 48.5 [3] | – | – | $48.4_{\pm 1.9}$ | 48.7 | $50.7_{\pm 0.2}$ | 51.0 | [16] | 6 |
| DART | Rouge-1 | – | 11.3 | 56.0 [3] | – | – | $45.5_{\pm 4.2}$ | 48.9 | $57.9_{\pm 1.6}$ | 59.2 | [11] | 7 |
| | Rouge-2 | – | 1.5 | 29.6 [3] | – | – | $25.0_{\pm 3.7}$ | 30.0 | $35.8_{\pm 1.0}$ | 36.2 | [11] | 7 |
| | Rouge-L | – | 3.2 | 48.5 [3] | – | – | $38.4_{\pm 3.8}$ | 43.4 | $48.5_{\pm 0.9}$ | 48.2 | [11] | 7 |
| E2ENLG | Rouge-1 | $72.6^a$ | 6.2 | 56.7 [3] | – | – | $44.8_{\pm 3.9}$ | 51.4 | $59.1_{\pm 1.3}$ | 59.7 | [12] | 9 |
| | Rouge-2 | $47.5^a$ | 2.5 | 31.4 [3] | – | – | $24.2_{\pm 3.6}$ | 30.1 | $33.2_{\pm 1.1}$ | 33.6 | [12] | 9 |
| | Rouge-L | $56.4^a$ | 4.9 | 41.1 [3] | – | – | $37.0_{\pm 3.5}$ | 42.4 | $44.9_{\pm 0.8}$ | 45.1 | [12] | 9 |
| WebNLG | Rouge-1 | $83.5^a$ | 13.9 | 68.3 [3] | – | – | $50.6_{\pm 4.7}$ | 57.7 | $68.5_{\pm 2.2}$ | 71.2 | [10] | 8 |
| | Rouge-2 | $63.6^a$ | 6.9 | 46.0 [3] | – | – | $29.8_{\pm 4.2}$ | 35.4 | $48.0_{\pm 1.5}$ | 49.8 | [10] | 8 |
| | Rouge-L | $71.0^a$ | 11.8 | 56.5 [3] | – | – | $43.4_{\pm 4.5}$ | 49.7 | $58.8_{\pm 1.1}$ | 60.2 | [10] | 8 |

Table 1: Results for translation and struct-to-text tasks. [k] indicates the number of few-shot exemplars. #t indicates the number of templates that FLAN is evaluated on. [a]T5-11B, [c]Edunov et al. (2018), [d]Durrani et al. (2014), [e]Wang et al. (2019b), [f]Sennrich et al. (2016), [g]Liu et al. (2020).

| | | | GLaM | | LaMDA-PT | | GPT-3 175B | | FLAN 137B zero-shot | | few-shot | | | |
|---|---|---|---|---|---|---|---|---|---|---|---|---|---|---|
| | Random Guess | Supervised Model | zero-shot | one-shot | zero-shot | few-shot [k] | zero-shot | few-shot [k] | average template | best dev template | average template | best dev template | [k] | #t |
| **NLI** | | | | | | | | | | | | | | |
| ANLI R1 | 33.3 | 57.4[b] | 40.9 | 42.4 | 39.6 | 39.0 [5] | 34.6 | 36.8 [50] | 47.7±1.4 | 46.4 | 44.2±2.3 | 47.9 | [6] | 8 |
| ANLI R2 | 33.3 | 48.3[b] | 38.2 | 40.0 | 39.9 | 37.5 [5] | 35.4 | 34.0 [50] | 43.9±1.3 | 44.0 | 41.6±1.4 | 41.1 | [6] | 8 |
| ANLI R3 | 33.3 | 43.5[b] | 40.9 | 40.8 | 39.3 | 40.7 [5] | 34.5 | 40.2 [50] | 47.0±1.3 | 48.5 | 42.8±2.2 | 46.8 | [6] | 8 |
| CB | 33.3 | 93.6[a] | 33.9 | 73.2 | 42.9 | 34.4 [5] | 46.4 | 82.1 [32] | 64.1±14.7 | 83.9 | 82.6±4.4 | 82.1 | [7] | 10 |
| MNLI-m | 33.3 | 92.2[a] | – | – | 35.7 | 43.7 [5] | – | – | 51.1±6.2 | 61.2 | 60.8±3.7 | 63.5 | [10] | 10 |
| MNLI-mm | 33.3 | 91.9[a] | – | – | 37.0 | 43.8 [5] | – | – | 51.0±6.5 | 62.4 | 61.0±3.5 | 63.5 | [10] | 10 |
| QNLI | 50.0 | 96.9[a] | – | – | 50.6 | 55.7 [5] | – | – | 59.6±4.9 | 66.4 | 62.0±1.7 | 63.3 | [12] | 9 |
| RTE | 50.0 | 92.5[a] | 68.8 | 71.5 | 73.3 | 70.8 [5] | 63.5 | 72.9 [32] | 78.3±7.9 | 84.1 | 79.9±6.9 | 84.5 | [8] | 10 |
| SNLI | 33.3 | 91.3[b] | – | – | 33.3 | 54.7 [5] | – | – | 43.0±7.4 | 53.4 | 62.3±2.4 | 65.6 | [15] | 9 |
| WNLI | 50.0 | 94.5[a] | – | – | 56.3 | 64.8 [5] | – | – | 61.0±10.6 | 74.6 | 55.4±11.0 | 70.4 | [14] | 10 |
| **READING COMP.** | | | | | | | | | | | | | | |
| BoolQ | 50.0 | 91.2[a] | 83.0 | 82.8 | 81.0 | 80.0 [1] | 60.5 | 77.5 [32] | 80.2±3.1 | 82.9 | 83.6±0.8 | 84.6 | [4] | 9 |
| DROP | – | 80.5[b] | 54.9 | 55.2 | 3.8 | 10.3 [1] | 23.6[†] | 36.5 [20] | 21.9±0.9 | 22.7 | 22.3±1.1 | 23.9 | [2] | 7 |
| MultiRC | – | 88.1[a] | 45.1 | 62.0 | 60.0 | 59.6 [5] | 72.9 | 74.8 [32] | 74.5±3.7 | 77.5 | 69.2±3.2 | 72.1 | [1] | 8 |
| OBQA | 25.0 | 85.4[a] | 53.0 | 55.2 | 41.8 | 50.6 [10] | 57.6 | 65.4 [100] | 77.4±1.3 | 78.4 | 77.2±1.3 | 78.2 | [16] | 7 |
| SQuADv1 | – | 96.2[a] | – | – | 22.7 | 50.2 [3] | – | – | 79.5±1.6 | 80.1 | 82.1±0.5 | 82.7 | [4] | 8 |
| SQuADv2 | – | 83.4[b] | 68.3 | 70.0 | 11.1 | 34.9 [3] | 59.5[†] | 69.8 [16] | 40.9±1.8 | 44.2 | 40.8±0.9 | 43.1 | [3] | 10 |
| **CLOSED-BOOK QA** | | | | | | | | | | | | | | |
| ARC-c | 25.0 | 81.1[a] | 48.2 | 50.3 | 42.0 | 49.4 [10] | 51.4 | 51.5 [50] | 61.7±1.4 | 63.1 | 63.7±0.6 | 63.8 | [13] | 7 |
| ARC-e | 25.0 | 92.6[a] | 71.9 | 76.6 | 76.4 | 80.9 [10] | 68.8 | 70.1 [50] | 79.5±0.8 | 79.6 | 80.5±0.5 | 80.7 | [14] | 7 |
| NQ | – | 36.6[a] | 21.5 | 23.9 | 3.2 | 22.1 [5] | 14.6 | 29.9 [64] | 18.6±2.7 | 20.7 | 27.2±0.5 | 27.6 | [16] | 10 |
| TQA (wiki) | – | 60.5[a] | 68.8 | 71.5 | 21.9 | 63.3 [10] | 64.3 | 71.2 [64] | 66.5±2.6 | 68.1 | 66.5±1.0 | 67.3 | [16] | 10 |
| TQA (tfds-dev) | – | 51.0[a] | – | – | 18.4 | 55.1 [10] | – | – – | 55.0±2.3 | 56.7 | 57.2±0.6 | 57.8 | [16] | 10 |
| **COMMONSENSE** | | | | | | | | | | | | | | |
| COPA | 50.0 | 94.8[a] | 90.0 | 92.0 | 90.0 | 89.0 [10] | 91.0 | 92.0 [32] | 90.6±2.0 | 91.0 | 88.5±3.8 | 87.0 | [16] | 8 |
| HellaSwag | 25.0 | 47.3[b] | 77.1 | 76.8 | 57.0 | 58.8 [10] | 78.9 | 79.3 [20] | 56.4±0.5 | 56.7 | 59.4±0.2 | 59.2 | [3] | 8 |
| PIQA | 50.0 | 66.8[b] | 80.4 | 81.4 | 80.3* | 80.2* [10] | 81.0 | 82.3 [50] | 80.9*±0.8 | 80.5* | 82.1*±0.3 | 81.7* | [10] | 8 |
| StoryCloze | 50.0 | 89.2[b] | 82.5 | 84.0 | 79.5 | 83.7 [10] | 83.2 | 87.7 [70] | 92.2±1.3 | 93.4 | 93.3±0.9 | 94.7 | [10] | 8 |
| **SENTIMENT** | | | | | | | | | | | | | | |
| IMDB | 50.0 | 95.5[b] | – | – | 76.9 | 83.3 [1] | – | – | 94.1±0.4 | 94.3 | 94.8±0.3 | 95.0 | [2] | 7 |
| Sent140 | 50.0 | 87.0[b] | – | – | 41.4 | 63.3 [5] | – | – | 69.9±2.5 | 73.5 | 68.7±1.2 | 69.3 | [16] | 6 |
| SST-2 | 50.0 | 97.5[a] | – | – | 51.0 | 92.3 [5] | 71.6 | 95.6 [8] | 92.6±1.7 | 94.6 | 94.4±0.8 | 94.6 | [16] | 8 |
| Yelp | 50.0 | 98.1[b] | – | – | 84.7 | 89.6 [3] | – | – | 97.8±0.2 | 98.1 | 97.9±0.1 | 98.0 | [4] | 7 |
| **PARAPHRASE** | | | | | | | | | | | | | | |
| MRPC | 50.0 | 90.4[a] | – | – | 53.7 | 64.0 [5] | – | – | 69.1±1.3 | 69.1 | 67.5±1.7 | 67.2 | [10] | 10 |
| QQP | 50.0 | 90.6[a] | – | – | 34.9 | 58.9 [3] | – | – | 72.1±6.8 | 75.9 | 73.5±2.9 | 75.9 | [16] | 7 |
| PAWS Wiki | 50.0 | 91.9[a] | – | – | 45.5 | 53.5 [5] | – | – | 61.5±6.5 | 69.4 | 66.2±0.9 | 70.2 | [10] | 10 |
| **COREFERENCE** | | | | | | | | | | | | | | |
| DPR | 50.0 | 84.8[b] | – | – | 54.6 | 57.3 [5] | – | – | 60.3±3.5 | 66.8 | 62.4±1.6 | 63.3 | [16] | 10 |
| Winogrande | 50.0 | 65.8[b] | 73.4 | 73.0 | 68.3 | 68.4 [10] | 70.2 | 77.7 [50] | 67.3±2.5 | 71.2 | 72.3±0.9 | 72.8 | [16] | 10 |
| WSC273 | 50.0 | 70.0[b] | 86.8 | 83.9 | 81.0 | 61.5 [5] | 88.3 | 88.5 [32] | 80.8±3.7 | – | – ± – | – | [–] | 10 |
| **READ. COMP. W/ COMMONSENSE** | | | | | | | | | | | | | | |
| CosmosQA | 25.0 | 67.1[b] | – | – | 34.1 | 33.8 [5] | – | – | 58.4±1.3 | 60.6 | 56.7±1.3 | 56.0 | [5] | 8 |
| ReCoRD | – | 93.4[a] | 90.3 | 90.3 | 87.8* | 87.6* [1] | 90.2 | 89.0 [32] | 67.8*±3.0 | 72.5* | 77.0*±2.0 | 79.0* | [1] | 10 |

Table 2: Results for eight NLU task clusters. All values shown are for accuracy (or exact match) except DROP, MultiRC, and SQuAD v1 and v2, which are F1. [k] indicates the number of few-shot exemplars. #t indicates the number of templates that FLAN is evaluated on.[a]T5-11B, [b]BERT-large. *see data contamination (Appendix C). WSC273 (Levesque et al., 2012) does not have training or validation sets, and so we do not compute few-shot results for FLAN. For Trivia QA (TQA), we report exact match (EM) on both the wikipedia subset of the dev set to compare with GPT-3, as well as the full TFDS dev set.

## B    FURTHER ABLATION STUDIES AND ANALYSIS

### B.1    DATASETS PER TASK CLUSTER & TEMPLATES PER DATASET

Our primary hypothesis is that instruction tuning on a diverse set of tasks improves performance on unseen tasks. §4.1 showed that adding more task clusters improves performance; here, we further explore whether adding additional datasets improves performance when the number of task clusters is held constant. We use the same split as in §4.1, where the NLI, commonsense reasoning, and closed-book QA clusters are held-out, and seven other task clusters remain for instruction tuning. For these seven task clusters, we instruction tune models using just one dataset per task cluster and using four datasets per task cluster (for task clusters that did not have four tasks, we just used all available tasks). In addition, we simultaneously explore the role of the number of instruction templates per dataset; as mentioned in §2.1, for each dataset we manually composed ten instructional templates for instruction tuning. Here, we instruction tune models using 1, 4, and 10 templates per dataset.

Figure 11 shows these results. Using more datasets per cluster improved performance by almost 10% on average across the three held-out clusters. Using more templates per dataset, however, had a comparatively negligible effect on performance when there was one task per cluster, which disappeared when there were four tasks per cluster. The small effect of templates is striking given our original motivation that composing ten templates per task would mitigate overfitting to any particular template. This results serves to underscore, however, the unpredictability of finetuning large language models, as one hypothesis is that models at such scale do not easily overfit to a finetuning single task.

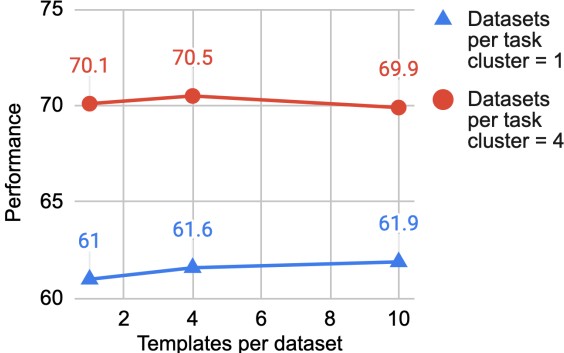

Figure 11: Effect of datasets per task cluster and templates per dataset on performance on three held-out clusters: NLI, commonsense reasoning, and closed-book QA. Adding more datasets per task cluster substantially improves performance. Using more templates per dataset, however, only had a very small effect on performance, which disappeared when there were sufficient dataset per task cluster.

### B.2    ROLE OF INSTRUCTIONS DURING FINETUNING

The per-cluster results for the ablation study from §4.3 are shown in Table 3.

### B.3    FURTHER ANALYSIS: INSTRUCTION TUNING FACILITATES PROMPT TUNING

The per-dataset results for the analysis in §4.5 are given in Table 4. As the above tasks are all classification, further work in this direction might include tasks such as summarization or question answering, or try to finetune the model using the supervised datasets.

## C    DATA CONTAMINATION ANALYSIS

One reasonable concern is that since the pretraining corpus of FLAN has more than 2 trillion tokens, it is possible that examples from a given evaluation dataset may have already been seen verbatim by the model during pre-training, hence inflating the performance of our purported zero-shot model. To this end, like GPT-3 (Brown et al., 2020), we perform post-hoc data contamination analysis to

| Finetuning prompt | Inference prompt | Zero-shot performance on unseen task cluster | | | | |
|---|---|---|---|---|---|---|
| | | NLI | Read. Comp. | Closed-Book QA | Translation | Four-Task Average |
| Natural instructions (= FLAN) | Natural instructions | 56.2 | 77.4 | 56.6 | 30.7 | 55.2 |
| No template | Natural instructions | 50.5 | 58.2 | 25.5 | 15.0 | 37.3 |
| Task/dataset name | Natural instructions | 52.8 | 63.0 | 44.8 | 25.9 | 46.6 |
| Task/dataset name | Task/dataset name | 60.2 | 64.9 | 40.8 | 21.9 | 47.0 |

Table 3: Ablation study result using models where instructions are removed from the finetuning process. In "no template," only inputs and outputs are given, which does not distinguish among tasks during multi-task finetuning. In "task/dataset name", inputs during multi-task finetuning are prepended with the name of the task and dataset (e.g., *"[Translation: WMT'14 to French] The dog runs"*) NLI datasets: ANLI R1–R3, CB, and RTE; reading comprehension datasets: BoolQ, MultiRC, and OpenbookQA; closed-book QA datasets: ARC-c, ARC-e, NQ, and TQA; translation datasets: WMT'14 Fr↔En, WMT'16 De↔En, and WMT'16 Ro↔En. Notably, training with task/dataset name achieved a high NLI score largely because it achieved a score of 83.9 on the CB dataset, for which the validation set only has 56 examples (FLAN also gets 83.9 with the best dev template, but the average template was only 64.1).

| | Prompt tuning train. examples | PROMPT TUNING ANALYSIS | | | | | | | |
|---|---|---|---|---|---|---|---|---|---|
| | | BoolQ acc. | CB acc. | CoPA acc. | MultiRC F1 | ReCoRD acc. | RTE acc. | WiC acc. | WSC acc. |
| LaMDA-PT | 32 | 55.5 | 55.4 | 87.0 | 65.4 | 78.0 | 52.4 | 51.6 | 65.4 |
| FLAN | | 77.5 | 87.5 | 91.0 | 76.8 | 80.8 | 83.0 | 57.8 | 70.2 |
| LaMDA-PT | full dataset | 82.8 | 87.5 | 90.0 | 78.6 | 84.8 | 82.0 | 54.9 | 72.7 |
| FLAN | | 86.3 | 98.2 | 94.0 | 83.4 | 85.1 | 91.7 | 74.0 | 86.5 |

Table 4: FLAN (instruction tuning) responds better to continuous inputs attained via prompt tuning than LaMDA-PT (no instruction tuning). When prompt tuning on a given dataset, no tasks from the same cluster as that dataset were seen during instruction tuning.

investigate whether the performance of the model is in fact inflated by evaluating on examples that occurred in the pretraining dataset.

Our data contamination procedure follows the setup of Brown et al. (2020), which, for each benchmark, produces a "clean" version that removes all potentially leaked examples, defined as examples for which any $n$-gram ($n$ varies per dataset but is roughly 13) overlapped with anything in the pretraining corpus. We use the same $n$ per dataset as Brown et al. (2020) and also split on spaces. We then evaluate our model on this clean subset, comparing against model performance on the original dataset (clean + dirty). Lower performance on the clean subset would suggest that data contamination leads to inflated results.

Figure 12 summarizes these results, with the exact numbers given in Table 5. We see several trends very similar to those in the GPT-3 paper: (1) many datasets had a substantial number of examples that overlapped with the pretraining data, (2) across all datasets, we do not see a correlation that evaluating on clean data does worse than evaluating on the total dataset, and (3) as datasets had fewer clean examples, there was higher variance in the percent change in performance (likely due to a smaller number of clean examples).

Like GPT-3, we also found that DROP and SQuADv2 had almost total overlap with the pretraining data. We follow their procedure of manually inspecting the data, and find that most overlapping $n$-grams were only in the contexts of examples (99.6% for DROP and 97.2% for SQuADv2). Overlaps never occurred in both the question and answer for DROP, and only occurred for both the question and answer for SQuADv2 in 5 of the 11,153 evaluation examples. Hence, for these two datasets, the

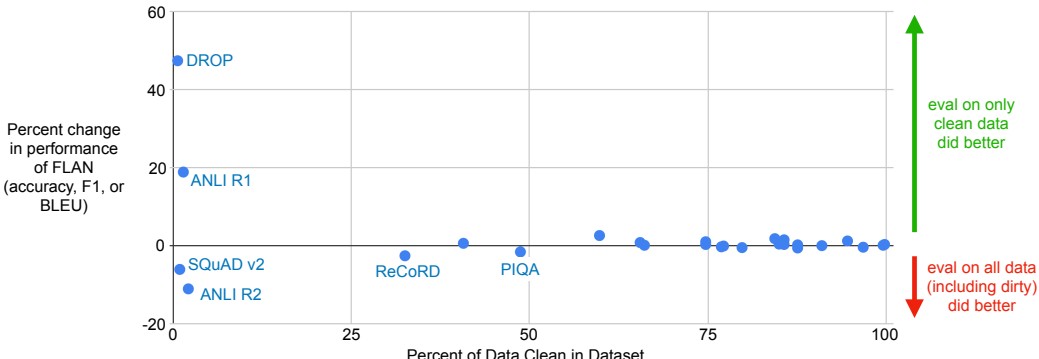

Figure 12: Like GPT-3, we also measured performance on cleaned versions of our datasets, which had high confidence to be unseen in the pretraining data of FLAN. We do not see a correlation that FLAN performed better on evaluation sets for which examples occurred more often in the pretraining data. When the percent of clean data is very small, there are fewer examples for computing the clean performance, which leads to high variance.

model gains only background information and cannot memorize the answer to any specific questions (aside from the five examples in SQuADv2).

ANLI R1 and R2 (Nie et al., 2020) also had almost complete data contamination, to a much higher degree than GPT-3. Upon further inspection, we see that most overlaps occur in example contexts and not hypotheses (97.3% for ANLI R1 and 98.2% for ANLI R2). As ANLI R1 and R2 are based entirely from Wikipedia examples (R3 is not), we posit that this higher degree of contamination in our pretraining dataset compared with GPT-3's is potentially due to using a more-recent version of Wikipedia that includes the contexts used in ANLI R1 and R2 (which were collected in 2019). Because seeing a particular context in pretraining does not help with the NLI task given a new, unseen sentence, we think it is unlikely that these overlaps affected performance on the two datasets.

Of the remaining datasets, only ReCoRD and PIQA had a clean subset performance that was lower than the overall evaluation set performance by more than 1%. These two datasets are language modeling (i.e., "what's the best continuation of this sentence?"), and so it is more likely compared with previous tasks that seeing a complete sentence in the pretraining data could help the model predict the right answer in downstream evaluations. For PIQA, both the goal and solution had overlaps in 93 of the 1,838 evaluation examples, and for ReCoRD, the query had overlaps in 2,320 of the 10,000 training examples. We hence mark these results with an asterisk * in Table 2. Brown et al. (2020) also reported substantial contamination rates for these two datasets (61% dirty for ReCoRD and 29% for PIQA), and also mark PIQA with an asterisk.

As this overlap analysis follows that performed in Brown et al. (2020), we reiterate the same caveats: the conservative nature of our $n$-gram matching procedure likely introduces additional false positives; there are no guarantees that the clean subset is drawn from the same distribution as the overall subset; and, accurately detecting test contamination is a relatively new research area without established best practices. Moreover, as our pretraining corpus is almost five times larger than that used for GPT-3 (which was 500B tokens), it is possible that there are more false positives in detecting dirty data.

| Dataset | Metric | Total count | Total acc/F1/BLEU | Clean count | Clean acc/F1/BLEU | % clean | % Diff (clean − overall) |
|---|---|---|---|---|---|---|---|
| DROP | F1 | 9,536 | 22.4 | 61 | 33.0 | 0.6 | 47.4 |
| SQuADv2 | F1 | 11,873 | 41.3 | 106 | 38.7 | 0.9 | -6.2 |
| ANLI R1 | acc | 1,000 | 48.1 | 14 | 57.1 | 1.4 | 18.8 |
| ANLI R2 | acc | 1,000 | 42.9 | 21 | 38.1 | 2.1 | -11.2 |
| ReCoRD | acc | 10,000 | 4.6 | 3,203 | 4.5 | 32.0 | -2.7 |
| MultiRC | acc | 4,848 | 75.4 | 1,972 | 75.7 | 40.7 | 0.5 |
| PIQA | acc | 1,838 | 23.7 | 896 | 23.3 | 48.7 | -1.7 |
| ANLI R3 | acc | 1,200 | 44.2 | 718 | 45.3 | 59.8 | 2.5 |
| HellaSwag | acc | 10,042 | 28.5 | 6,578 | 28.7 | 65.5 | 0.7 |
| RTE | acc | 2,77 | 84.1 | 183 | 84.2 | 66.1 | 0.0 |
| WMT'14 En→Fr | BLEU | 3,003 | 31.3 | 2,243 | 31.5 | 74.7 | 0.9 |
| WMT'14 Fr→En | BLEU | 3,003 | 34.0 | 2,243 | 34.1 | 74.7 | 0.2 |
| BoolQ | acc | 3,270 | 76.5 | 2,515 | 76.3 | 76.9 | -0.4 |
| TQA (tfds-dev) | F1 | 11,313 | 62.2 | 8,731 | 62.0 | 77.2 | -0.2 |
| ARC Easy | acc | 2,365 | 79.5 | 1,888 | 79.0 | 79.8 | -0.6 |
| ARC Challenge | acc | 1,165 | 63.1 | 983 | 64.2 | 84.4 | 1.7 |
| OpenbookQA | acc | 500 | 74.6 | 425 | 74.8 | 85.0 | 0.3 |
| WMT'16 En→De | BLEU | 2,999 | 22.7 | 2,569 | 23.0 | 85.7 | 1.4 |
| WMT'16 De→En | BLEU | 2,999 | 38.6 | 2,569 | 38.7 | 85.7 | 0.2 |
| WMT'16 En→Ro | BLEU | 1,999 | 15.5 | 1,752 | 15.4 | 87.6 | -0.7 |
| WMT'16 Ro→En | BLEU | 1,999 | 36.7 | 1,752 | 36.8 | 87.6 | 0.1 |
| COPA | acc | 100 | 88.0 | 91 | 87.9 | 91.0 | -0.1 |
| CB | acc | 56 | 41.1 | 53 | 41.5 | 94.6 | 1.1 |
| NQ | F1 | 3,610 | 24.5 | 3,495 | 24.3 | 96.8 | -0.5 |
| StoryCloze | acc | 1,871 | 92.1 | 1,864 | 92.1 | 99.6 | 0.0 |
| Winogrande | acc | 1,267 | 39.4 | 1,265 | 39.4 | 99.8 | 0.2 |

Table 5: Overlap statistics for the subset of datasets that are also used in GPT-3, sorted from dirtiest to cleanest. An evaluation example was dirty if it had any $n$-gram collision with the pretraining corpus. We computed these results for FLAN's performance using only a single template for each dataset, so they differ slightly compared with the results for average performance over all templates.

## D EXTENDED RELATED WORK

### D.1 LANGUAGE MODELS AND MULTI-TASK LEARNING

Our work is broadly inspired by a long line of prior work on language models for NLP applications (Dai & Le, 2015; Peters et al., 2018; Howard & Ruder, 2018; Radford et al., 2018; 2019, *inter alia*). Instruction tuning can be seen as a formulation of multitask learning (MTL), which is an established area within deep learning (Collobert et al., 2011; Luong et al., 2016; Ruder, 2017; Velay & Daniel, 2018; Clark et al., 2019b; Liu et al., 2019b, *inter alia*)—see Worsham & Kalita (2020) for a recent survey on MTL for NLP. Differing from prior MTL work which focuses on performance improvements across training tasks (Raffel et al., 2020; Aghajanyan et al., 2021) or to new domains (Axelrod et al., 2011), our work is motivated by improving zero-shot generalization to tasks that were not seen in training.

### D.2 ZERO-SHOT LEARNING AND META-LEARNING

Our work also falls in the well-established category of zero-shot learning, which has historically been used to refer to classifying instances among a set of unseen categories (Lampert et al., 2009; Romera-Paredes & Torr, 2015; Srivastava et al., 2018; Yin et al., 2019, *inter alia*). In NLP, zero-shot learning work also includes translating between unseen language pairs (Johnson et al., 2017; Pham et al., 2019), language modeling on unseen languages (Lauscher et al., 2020), as well as various NLP applications (Liu et al., 2019a; Corazza et al., 2020; Wang et al., 2021). Most recently, the emergent ability of language models (Brown et al., 2020) has led to increased interest in how models generalize to unseen tasks, the definition of zero-shot learning used in our paper. In addition, meta-learning (Finn et al., 2017; Vanschoren, 2018, *inter alia*) also broadly tries to train models that adapt quickly to unseen tasks, typically based on a few examples.

### D.3 PROMPTING

Instruction tuning leverages the intuition that language models at scale contain substantial world knowledge and can perform a range of NLP tasks (Brown et al., 2020, see also Bommasani et al. (2021)). Another line of work that shares this goal prompts models with continuous inputs optimized via backpropagation to substantially improve performance (Li & Liang, 2021; Lester et al., 2021; Qin & Eisner, 2021), as well as work that prompts models to produce specialized outputs (Wei et al., 2022). Although the success of these approaches depends heavily on model scale (Lester et al., 2021), for which large models can be costly to serve, the ability of a single large model to perform many tasks slightly eases this burden. As shown by our experiments in §4.5, prompt tuning is an orthogonal method for which instruction tuning can additionally improve performance. Reif et al. (2021) is similar to our work in that they also use related tasks to improve zero-shot learning, though they differ by only using related tasks in the context (and not finetuning), and focus on the application of text style transfer.

Our work shares similar motivations with prompting in that we use inference-time text interactions to prompt a single model, without creating separate checkpoints for each task. Whereas prompting work such as GPT-3 uses prompt engineering to write prompts that intentionally mimic text that is likely to be seen during pretraining (e.g., for MultiRC GPT-3 tries a prompt that mimics a test with an answer key), we hope that finetuning models to respond to natural language instructions instead of completing a sentence will make such large models more accessible to non-technical users.

### D.4 FINETUNING LARGE LANGUAGE MODELS

Finetuning pretrained language models is a well-established method in NLP, with much of the work so far occurring on models in the range of 100M to 10B parameters (Dai & Le, 2015; Devlin et al., 2019; Raffel et al., 2020; Lewis et al., 2020, *inter alia*). For models of O(100B) parameters, recent work has finetuned task-specific models for program synthesis (Austin et al., 2021; Chen et al., 2021), summarization (Wu et al., 2021), as well as improved bias and fairness behavior (Solaiman & Dennison, 2021). In addition to the traditional "dense" models, sparse mixture of experts (MoE) models of up to more than 1T parameters have been trained and finetuned (Lepikhin et al., 2020; Fedus

et al., 2021). Compared with this prior work that finetunes and evaluates on the same downstream task, our setup studies the effect of instruction tuning on ability to perform unseen tasks.

## D.5 MULTI-TASK QUESTION ANSWERING

The instructions we use for instruction tuning are similar to QA-based task formulation research, which aims to unify NLP tasks by casting them as question-answering over a context. For instance, McCann et al. (2018) cast ten NLP tasks as QA and train a model on a collection of tasks formulated with natural language prompts; they report transfer learning gains on finetuning tasks as well as zero-shot domain adaptation results on SNLI (Bowman et al., 2015) and Amazon/Yelp Reviews (Kotzias et al., 2015). While McCann et al. (2018) does not leverage unsupervised pre-training and only reports zero-shot transfer to unseen domains, our work uses a pretrained LM and focuses on zero-shot performance on unseen task clusters. UnifiedQA (Khashabi et al., 2020) shows similar transfer learning gains as McCann et al. (2018) across 20 datasets and reports good generalization to unseen tasks across four types of QA. Focusing on binary text classification, Zhong et al. (2021) finetune T5-770M on 43 tasks phrased as yes/no questions and study the zero-shot performance on unseen tasks. In comparison, our paper is much larger in scope, empirically demonstrating the idea on a wide range of tasks with a much larger model. Other work has used QA-based task formulation for more-targeted applications including semantic role labeling (He et al., 2015), relation extraction (Levy et al., 2017), coreference resolution (Wu et al., 2020) and named entity recognition (Li et al., 2020) as question answering.

## D.6 INSTRUCTIONS-BASED NLP

Recent improvements in the capabilities of language models have led to increased interest in a nascent area of instructions-based NLP (Goldwasser & Roth, 2014, and see McCarthy (1960)). Schick & Schütze (2021) (also see Gao et al., 2021; Tam et al., 2021) use task descriptions in cloze-style phrases to help language models assign soft labels for few-shot and semi-supervised learning, though this line of work finetunes new checkpoints for each downstream task. Efrat & Levy (2020) evaluated GPT-2 (Radford et al., 2019) on simple tasks ranging from retrieving the $n$th word of a sentence to generating examples for SQuAD, concluding that GPT-2 performs poorly across all tasks.

In terms of the setup of finetuning on a large number of tasks and evaluating on unseen tasks, two recent papers are similar to ours. Mishra et al. (2021) finetune BART (Lewis et al., 2020) using instructions and few-shot examples for tasks such as question answering, text classification, and text modification, and find that this few-shot finetuning with instructions improves performance on unseen tasks. Ye et al. (2021) introduce a setup for cross-task few-shot learning, finding that multi-task meta-learning using MAML (Finn et al., 2017) improves the few-shot capabilities of BART on unseen downstream tasks. Our work differs from these two papers in that we focus on zero-shot learning, for which we observe the crucial importance of model scale (FLAN is 1,000x larger than BART-base).

Perhaps the papers most related to ours are the recent Sanh et al. (2021) and Min et al. (2021), which were released after our initial preprint. Min et al. (2021) finetunes GPT-2 Large (770M parameters) to be a few-shot learner, which is the same approach as our experiment in Section 4.3. Similar to our conclusions, they also observe that including few-shot exemplars and instruction tuning are complementary ways to improve performance. Sanh et al. (2021) propose to finetune T5-11B to respond to prompts, and they also report performance improvements on zero-shot learning. These two papers and our work all study finetuning with instructions, but, as noted by Min et al. (2021), it is hard to directly compare results, due to differing model sizes, model types (decoder-only vs encoder-decoder), pretraining data, task mixtures, and type of instructions (Sanh et al. (2021) say that their instructions are more diverse).

Finally, OpenAI has a model called InstructGPT (Ouyang et al., 2022). InstructGPT uses human anntations to guide desired model behavior, both via finetuning and reinforcement learning, finding that InstructGPT is preferred by human rathers compared with unmodified GPT-3.

## E   FREQUENTLY ASKED QUESTIONS

**How do the FLAN instructions differ from GPT-3 or T5 prompts?**

GPT-3 prompting is done in a way such that the prompt looks like data that the model has been pretrained on, and the model finishes the continuation. T5 prompts are mostly just a tag for the dataset, which would not work in the zero-shot setting. In contrast, the prompts that we use for FLAN are similar to what would be used to ask a human to perform the task.

For instance, given an input for an NLI task, these would be the prompts.

*T5 prompt:*
```
cb hypothesis:  At my age you will probably have learnt one lesson.
premise:  It's not certain how many lessons you'll learn by your
thirties.
```

*GPT-3 prompt:*
```
At my age you will probably have learnt one lesson.
question:  It's not certain how many lessons you'll learn by your
thirties.  true, false, or neither?  answer:
```

*FLAN prompt:*
```
Premise:  At my age you will probably have learnt one lesson.
Hypothesis:  It's not certain how many lessons you'll learn by your
thirties.
Does the premise entail the hypothesis?
```

So because FLAN prompts are formulated as responding to an instruction, they do not work well for pretrained language models without finetuning. Performance was near zero for most generation tasks. For instance, given the input *"'The dog runs.' Translate this sentence to French."*, LaMDA-PT continues with *"The dog runs after the cat"* instead of actually translating the sentence. Hence, we used the established GPT-3 prompts for our LaMDA-PT baselines.

**What are some limitations/failure cases of FLAN?**

While we qualitatively find that FLAN responds well to most tasks, it does fail on some simple tasks. For instance, as shown in Figure 22, FLAN fails at the very simple task of returning the second word in a sentence, and also incorrectly translates a question to Danish when asked to answer the question in Danish. Additional limitations include a context length of only 1024 tokens (which is not enough for most summarization tasks), and that the model was mostly trained on English data.

**Can FLAN be used when large amounts of training data are available?**

In this work, we focus on cross-task generalization to zero-shot tasks, but we also believe that instruction tuning could result in positive task transfer among seen tasks, depending on the mixture of tasks (though we leave this for future work). In §4.5, where we apply prompt tuning to the FLAN checkpoint, we see promising results that indicate positive task transfer in a supervised setting.

**Are the ten unique templates per dataset or per task cluster?**

The ten unique templates are for each dataset and not for a task cluster. This is because datasets in the same task cluster often differed slightly (e.g., *"is this movie review positive"* vs *"is this yelp review positive"*).

**In Figure 7A, why does the untuned LaMDA-PT model see worse performance with more parameters for reading comprehension and sentiment analysis?**

For context, Figure 7A is a check of correctness for Figure 7B. Figure 7A confirms that scale improves performance for tasks that were seen during instruction tuning, as expected. The untuned LaMDA-PT model performance in Figure 7A is shown just for completeness.

Nonetheless, the fact that scale does not always improve zero-shot performance of untuned LaMDA-PT is an interesting artifact. Initially, we were surprised, because Brown et al. (2020) shows that scale improves performance across a large number of tasks in aggregate.

It turns out that scale does not improve performance for certain tasks. This is especially true for zero-shot learning, and we think that this happens to be the case for the reading comprehension and sentiment analysis tasks we evaluate. The GPT-3 paper itself similarly reports that zero-shot performance on BoolQ and DROP decreases from 13B to 175B parameters. The GPT-3 paper does not show results on sentiment analysis, but Holtzman et al. (2021) find that zero-shot performance on SST-2 also gets worse from 13B to 175B parameters. Hence, this artifact is consistent across both GPT-3 and the models we use.

This artifact is certainly worth further study, but is outside the scope of instruction tuning. Ideally, we would have performed the Figure 7 ablation with cross-validation instead of a single split, which likely would have smoothed out that artifact.

## F QUALITATIVE EXAMPLES

This section shows qualitative examples of FLAN responding to various prompts.

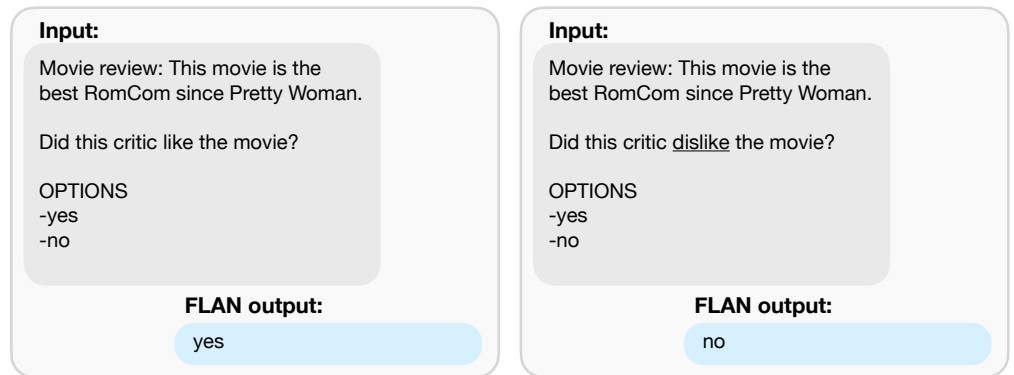

Figure 13: For sentiment analysis, FLAN changes the answer appropriately when the question is flipped.

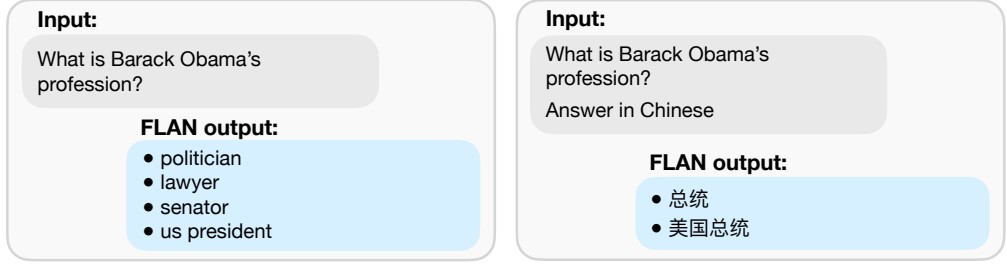

Figure 14: For question answering, FLAN can answer a question in another language when instructed to do so. Multiple FLAN outputs are generated via random sampling with a temperature of 0.9 and top $k$ of 40.

**Input:**

John is reading the paper "Finetuned Language Models are Zero-Shot Learners."

Change to past tense.

**FLAN output:**

John was reading the paper "Finetuned Language Models are Zero-Shot Learners."

**Input:**

John is reading the paper "Finetuned Language Models are Zero-Shot Learners."

Change to passive voice.

**FLAN output:**

The paper "Finetuned Language Models are Zero-Shot Learners" is being read by John.

**Input:**

John is reading the paper "Finetuned Language Models are Zero-Shot Learners."

Change to the verb to eat.

**FLAN output:**

John is eating the paper "Finetuned Language Models are Zero-Shot Learners."

**Input:**

John is reading the paper "Finetuned Language Models are Zero-Shot Learners."

Change the verb to eat and the subject to Noam Chomsky.

**FLAN output:**

Noam Chomsky is eating the paper "Finetuned Language Models are Zero-Shot Learners."

Figure 15: FLAN can perform zero-shot rule-based manipulations.

**Input:**

Recommend fun activities to do on a rainy night in Seattle.

**FLAN output:**

- Go to the pacific science center
- Go out to eat
- Go to a comedy club
- Build a blanket fort
- Go to the Space Needle
- Go to underground tours
- Find a karaoke bar

**Input:**

Recommend activities to do on a sunny weekend in Copenhagen.

**FLAN output:**

- Visit Christiania
- Copenhagen Zoo
- Visit the Tivoli Gardens
- Take a canal tour
- Museums
- Visit The Little Mermaid Statue
- Visit Copenhagen Zoo
- Visit the Danish Maritime Museum

Figure 16: FLAN can make zero-shot recommendations. Multiple FLAN outputs are generated via random sampling with a temperature of 0.9 and top $k$ of 40.

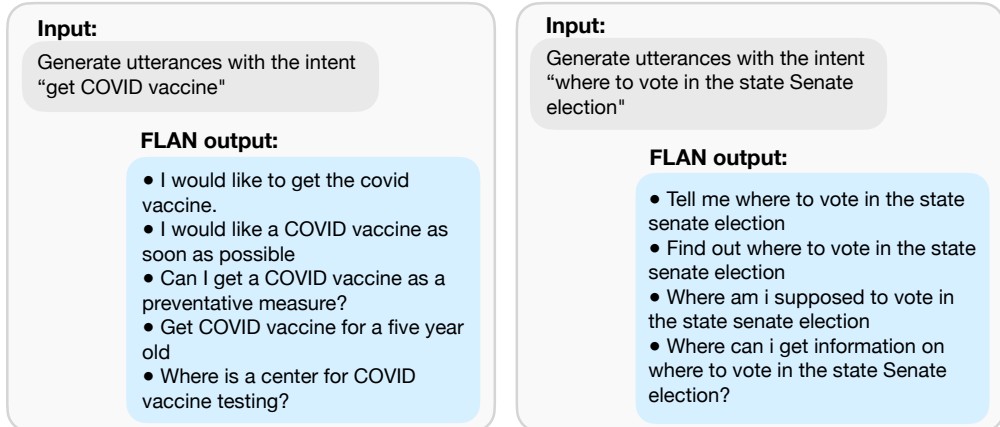

Figure 17: FLAN can be used in a zero-shot fashion to generate data, such as utterances that are consistent with a given intent. Multiple FLAN outputs are generated via random sampling with a temperature of 0.9 and top $k$ of 40.

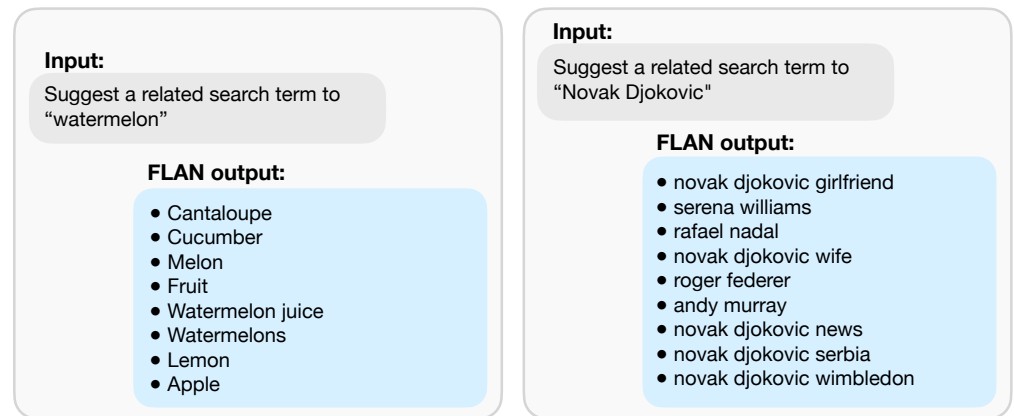

Figure 18: FLAN can be used for zero-shot query expansion. Multiple FLAN outputs are generated via random sampling with a temperature of 0.9 and top $k$ of 40.

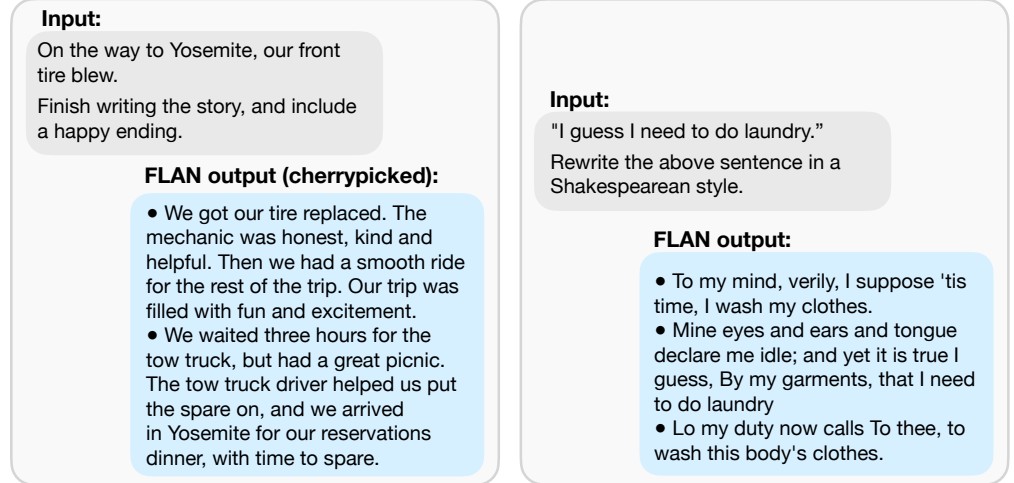

Figure 19: FLAN can perform zero-shot tasks relevant to assisted-writing applications. Multiple FLAN outputs are generated via random sampling with a temperature of 0.9 and top $k$ of 40.

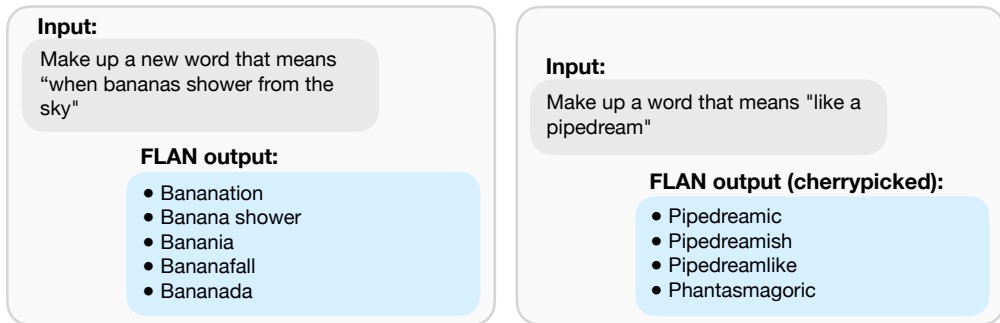

Figure 20: FLAN can be used for zero-shot word formation. Multiple FLAN outputs are generated via random sampling with a temperature of 0.9 and top $k$ of 40.

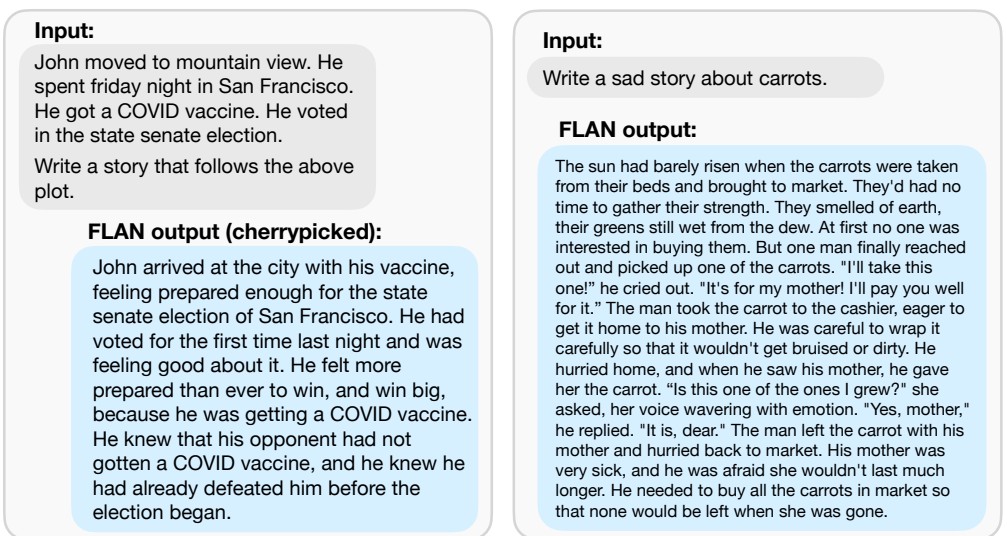

Figure 21: Open-ended generation tasks by FLAN. The carrot story was from sampling sixteen outputs with a minimum length of 150 and choosing the highest probability output.

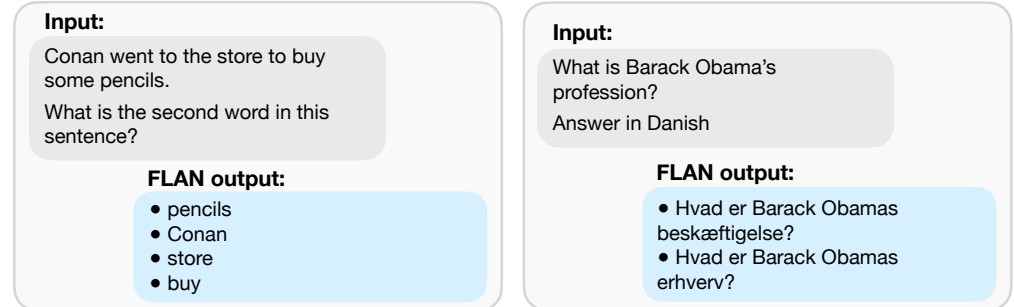

Figure 22: Example failure cases for FLAN. Left: FLAN fails to perform a simple task of returning the $n$th word. Right: FLAN translates a question instead of answering it. Multiple FLAN outputs are generated via random sampling with a temperature of 0.9 and top $k$ of 40.

## CHANGES FROM V4 TO V5

- Replaced the tables in the main figure with a figure, which takes up less space and focuses on zero-shot performance.
- Added GLaM 64B/64E as a baseline.
- Moved the ablation about the role of instructions, as well as prompt tuning, into the main paper (and condensed the figures).

## CHANGES TO V4 FROM V3

- We added a Frequently Asked Questions section (Appendix E).
- We added a section with qualitative examples (Appendix F).
- We added an additional ablation study on the role of instructions during finetuning (Appendix B.2).
- We updated the related work (Appendix D) with manuscripts posted on arxiv since our initial preprint.

## CHANGES TO V3 FROM V2

- The number of tokens used in pretraining was corrected from 2.81T to 2.49T tokens.

## CHANGES TO V2 FROM V1

- We updated the terminology to "datasets" and "task clusters."
- We renamed the previous "open-domain QA" task cluster to "closed-book QA."
- We extended the related work section and moved it to the Appendix D, using a shorter version in the main body.
- We added FLAN and LaMDA-PT results for additional datasets for which GPT-3 results were not reported.
- For TriviaQA, v1 reported results on the tfds dev set of 11,313 examples. GPT-3 actually evaluates on the wikipedia dev set of 7,993 examples, so we ran an additional evaluation on that dev set in order to compare with GPT-3's performance. Zero-shot FLAN now beats zero-shot GPT-3 on that task (and therefore on 20 of 25 tasks). We still show the original result in Table 2, though there is no GPT-3 result to compare with.
- We moved commonsense reasoning and coreference resolution from the main body to the Appendix.
- We moved prompt tuning from the main body to §4.5.
- We added data contamination analysis (Appendix C).
- We added few-shot instruction tuning (§4.4).
- We cited additional datasets in Appendix G.
- The number of tokens used in pretraining was corrected from 2.81T to 2.49T tokens.

## G    TASKS AND DATASETS

This appendix further details the datasets that we use in this paper. We group datasets into one of the following task clusters:

- **Natural language inference** concerns how two sentences relate, typically asking, given a first sentence, whether a second sentence is true, false, or possibly true. We use the following datasets:
    1. ANLI (Nie et al., 2020)
    2. CB (De Marneffe et al., 2019)
    3. MNLI (Williams et al., 2018)
    4. QNLI (Rajpurkar et al., 2018)
    5. SNLI (Bowman et al., 2015)
    6. WNLI (Levesque et al., 2012)
    7. RTE (Dagan et al., 2005; Haim et al., 2006; Giampiccolo et al., 2007; Bentivogli et al., 2009)

- **Reading comprehension** tests the ability to answer a question when given a passage that contains the answer. We use the following datasets:
    1. BoolQ Clark et al. (2019a)
    2. DROP (Dua et al., 2019)
    3. MultiRC (Khashabi et al., 2018)
    4. OBQA (Mihaylov et al., 2018)
    5. SQuADv1 (Rajpurkar et al., 2016)
    6. SQuADv2 (Rajpurkar et al., 2018)

- **Commonsense reasoning** evaluates the ability to perform physical or scientific reasoning with an element of common sense. We use the following datasets:
    1. COPA (Roemmele et al., 2011)
    2. HellaSwag (Zellers et al., 2019)
    3. PiQA (Bisk et al., 2020)
    4. StoryCloze (Mostafazadeh et al., 2016)

- **Sentiment analysis** is a classic NLP task aims to understand whether a piece of text is positive or negative. We use the following datasets:
    1. IMDB (Maas et al., 2011)
    2. Sentiment140 (Go et al., 2009)
    3. SST-2 (Socher et al., 2013)
    4. Yelp (Fast.AI)

- **Closed-book QA** asks models to answer questions about the world without specific access to information that contains the answer. We use the following datasets:
    1. ARC (Clark et al., 2018)
    2. NQ (Lee et al., 2019; Kwiatkowski et al., 2019)
    3. TriviaQA Joshi et al. (2017)

- **Paraphrase detection** asks a model to determine whether two sentences are semantically equivalent.[4] We use the following datasets:
    1. MRPC (Dolan & Brockett, 2005)
    2. QQP (Wang et al., 2018, see)
    3. Paws Wiki (Zhang et al., 2019)

- **Coreference resolution** tests the ability to identify expressions of the same entity in some given text. We use the following datasets:
    1. DPR (Rahman & Ng, 2012)
    2. Winogrande (Sakaguchi et al., 2020)

---

[4]Although paraphrasing can be seen as positive entailment in both directions, it has been distinct from NLI in the academic literature.

    3. WSC273 (Levesque et al., 2012)

- **Reading comprehension with commonsense** combines elements of both reading comprehension with commonsense. We use the following datasets:

  1. CosmosQA (Huang et al., 2019)
  2. ReCoRD (Zhang et al., 2018)

- **Struct to text** tests the ability to describe some structured data using natural language. We use the following datasets:

  1. CommonGen (Lin et al., 2020)
  2. DART (Nan et al., 2021)
  3. E2ENLG (Dušek et al., 2019)
  4. WebNLG (Gardent et al., 2017)

- **Translation** is the task of translating text from one language into a different language. We use the following datasets:

  1. En–Fr from WMT'14 (Bojar et al., 2014)
  2. En–De, En–Tr, En–Cs, En–Fi, En–Ro, and En–Ru from WMT'16 (Bojar et al., 2016)
  3. En–Es from Paracrawl (Bañón et al., 2020)

- **Summarization** asks models to read a piece of text and generate an abbreviated summary of it. We use the following datasets:

  1. AESLC (Zhang & Tetreault, 2019)
  2. CNN-DM (See et al., 2017)
  3. Gigaword (Napoles et al., 2012)
  4. MultiNews (Fabbri et al., 2019)
  5. Newsroom (Grusky et al., 2018)
  6. Samsum (Gliwa et al., 2019)
  7. XSum (Narayan et al., 2018)
  8. AG News (Zhang et al., 2015)
  9. Opinion Abstracts - Rotten Tomatoes (Wang & Ling, 2016)
  10. Opinion Abstracts - iDebate (Wang & Ling, 2016)
  11. Wiki Lingua English (Ladhak et al., 2020)

- Additional datasets that we assign to a miscellaneous task cluster include:

  1. Conversational question-answering: QuAC (Choi et al., 2018) and CoQA (Reddy et al., 2019)
  2. Evaluating context-sentence word meanings: WiC (Pilehvar & Camacho-Collados, 2019)
  3. Question classification: TREC (Li & Roth, 2002; Hovy et al., 2001)
  4. Linguistic acceptability: CoLA (Warstadt et al., 2019)
  5. Math questions (Saxton et al., 2019)

For all tasks, our finetuning and evaluation code uses tensorflow datasets (TFDS) to load and process datasets. Regarding the number of training examples per dataset, we limited the training set size per dataset to 30,000 so that no dataset dominated the finetuning distribution. When a test set with labels was available in TFDS, we used it; otherwise, we used the TFDS validation set as our test set, splitting the training set into a train and dev set.

On the following pages, we show inputs and outputs for evaluation tasks where we compared with GPT-3. See the attached supplementary material for the templates for all other datasets.

## G.1 NATURAL LANGUAGE INFERENCE

---

**INPUT**

Joey Heindle (born 14 May 1993 in Munich) is a German singer. He is best known for winning the seventh season of the game show Ich bin ein Star – Holt mich hier raus! and finishing in 5th place in season 9 of Deutschland sucht den Superstar, despite universally negative reviews from the jury each week.

Based on the paragraph above can we conclude that "Joey Heindle was highly disliked by people on television."?

OPTIONS:
- Yes
- It's impossible to say
- No

**TARGET**

Yes

---

Table 6: Example input and target for Adversarial NLI (ANLI). ANLI (Nie et al., 2020) is a large-scale NLI benchmark with adversarial examples collected iteratively with a human and model in the loop. The task is to determine whether a hypothesis is entailed by a premise (entailment, not entailment, or impossible to say). There are three rounds, R1–R3. Of the three training sets with 16,946, 45,460, and 100,459 examples, we use 16,946, 30,000, and 30,000 for train and 200 from each of the three TFDS validation sets for dev. We use the TFDS "test" sets of 1,000, 1,000, and 1,200 examples as our test set for reporting numbers.

---

**INPUT**

A: so I watch the fish, you know. Whatever I can do to keep myself occupied. I like to have the TV on, because that usually keeps me, um, more occupied. It kind of takes the time away and I don't realize, that's really the only time I ever watch TV, is when I'm on the bike. and then usually after I'm done riding the bike, just to cool myself down, I usually take a walk, you know, and that just kind of uh, gets me, you know, to where I'm not quite as tired I guess. But it's definitely a task. B: You think so? A: I can't say that I really enjoy it.

Based on the paragraph above can we conclude that "she really enjoys it"?

OPTIONS:
- Yes
- No
- It's impossible to say

**TARGET**

No

---

Table 7: Example input and target for Commitment Bank (CB). CB (De Marneffe et al., 2019) is a corpus of texts in which a hypothesis is extracted from a premise, and the task is to determine whether the hypothesis is entailed by the premise (entailment, not entailment, or impossible to say). Of the training set with 250 examples, we use 200 for train and 50 for dev. We use the TFDS validation set of 56 examples as our test set for reporting numbers.

| |
|---|
| **INPUT** |
| After years of study, the Vatican's doctrinal congregation has sent church leaders a confidential document concluding that "sex-change" procedures do not change a person's gender in the eyes of the church. |
| Based on the paragraph above can we conclude that "Sex-change operations become more common."? |
| OPTIONS: |
| - yes |
| - no |
| **TARGET** |
| no |

Table 8: Example input and target for Recognizing Textual Entailment (RTE). RTE (Dagan et al., 2005; Haim et al., 2006; Giampiccolo et al., 2007; Bentivogli et al., 2009) asks whether a second sentence is entailed by a first (binary, either entailed or not entailed). Of the training set with 2490 examples, we use 2,290 for train and 200 for dev. We use the TFDS validation set of 277 examples as our test set for reporting numbers.

## G.2 READING COMPREHENSION

---

**INPUT**

There are four ways an individual can acquire Canadian citizenship: by birth on Canadian soil; by descent (being born to a Canadian parent); by grant (naturalization); and by adoption. Among them, only citizenship by birth is granted automatically with limited exceptions, while citizenship by descent or adoption is acquired automatically if the specified conditions have been met. Citizenship by grant, on the other hand, must be approved by the Minister of Immigration, Refugees and Citizenship.

Can we conclude that can i get canadian citizenship if my grandfather was canadian?

OPTIONS:
- no
- yes

**TARGET**

no

---

Table 9: Example input and target for Boolean Questions (BoolQ). BoolQ Clark et al. (2019a) asks a yes/no question based on a passage and a question. Of the training set with 9,427 examples, we use 9,227 for train and 200 for dev. We use the TFDS validation set of 3,270 examples as our test set for reporting numbers.

---

**INPUT**

Imagine you are standing in a farm field in central Illinois. The land is so flat you can see for miles and miles. On a clear day, you might see a grain silo 20 miles away. You might think to yourself, it sure is flat around here. If you drive one hundred miles to the south, the landscape changes. In southern Illinois, there are rolling hills. Why do you think this is? What could have caused these features? There are no big rivers that may have eroded and deposited this material. The ground is capable of supporting grass and trees, so wind erosion would not explain it. To answer the question, you need to go back 12,000 years. Around 12,000 years ago, a giant ice sheet covered much of the Midwest United States. Springfield, Illinois, was covered by over a mile of ice. Its hard to imagine a mile thick sheet of ice. The massive ice sheet, called a glacier, caused the features on the land you see today. Where did glaciers go? Where can you see them today? Glaciers are masses of flowing ice.

Question: "How big were the glaciers?"

Response: "One mile"

Does the response correctly answer the question?

OPTIONS:
- no
- yes

**TARGET**

yes

---

Table 10: Example input and target for Multi-Sentence Reading Comprehension (MultiRC). MultiRC Khashabi et al. (2018) asks an open-ended question given a paragraph that contains the answer. Of the training set with 27,243 examples, we use 27,043 for train and 200 for dev. We use the TFDS validation set of 4,848 examples as our test set for reporting numbers.

| INPUT |
|---|
| soil is a renewable resource for growing plants
A plant that needs to expand will be able to have an endless resource in

OPTIONS:
- dirt
- pesticides
- pay
- beans |
| **TARGET**
dirt |

Table 11: Example input and target for Openbook Question Answering (OBQA). OBQA (Mihaylov et al., 2018) asks 4-way multiple choice questions based facts. Of the training set with 4,957 examples, we use all for train and 200 in the TFDS validation set of 500 examples for dev. We use the TFDS test set of 500 examples as our test set for reporting numbers.

## G.3 COMMONSENSE REASONING

| |
|---|
| **INPUT** |
| I packed up my belongings. What is the cause? |
| |
| OPTIONS: |
| - I was hunting for a new apartment. |
| - I was moving out of my apartment. |
| **TARGET** |
| I was moving out of my apartment. |

Table 12: Example input and target for Choice of Plausible Alternatives (COPA). COPA (Roemmele et al., 2011) is a causal reasoning task that asks to infer either a cause of effect of a premise from two choices. Of the training set with 400 examples, we use 350 for train and 50 for dev. We use the TFDS validation set of 100 examples as our test set for reporting numbers.

| |
|---|
| **INPUT** |
| What happens next in this paragraph? |
| |
| Once the rope is inside the hook, he begins moving up the wall but shortly after he stops and begins talking. The male then begins talking about the clip again and goes back up the wall. as he |
| OPTIONS: |
| - progresses, there are hooks everywhere on the wall and when he gets near them, he puts his rope inside of it for support and safety. |
| - changes time, an instant replay of his initial move is shown a second time. |
| - continues to talk, another male speaks about the move and shows another closeup of the plex by the male. |
| - continues, other people start to arrive and begin to hang out with him as he makes a few parts of the rope. |
| **TARGET** |
| progresses, there are hooks everywhere on the wall and when he gets near them, he puts his rope inside of it for support and safety. |

Table 13: Example input and target for Commonsense Sentence Completion (HellaSwag). HellaSwag (Zellers et al., 2019) tests for sentence completion that requires common sense, asking for the most probable ending given four contexts. Of the training set with 39,905 examples, we use 30,000 for train and 200 for dev. We use the TFDS validation set of 10,042 examples as our test set for reporting numbers.

---

**INPUT**
Here is a goal: Remove smell from garbage disposal.

How would you accomplish this goal?

OPTIONS:
- Create soda ice cubes and grind through disposal.
- Create vinegar ice cubes and grind through disposal.

**TARGET**
Create vinegar ice cubes and grind through disposal.

---

Table 14: Example input and target for Physical Question Answering (PiQA). PiQA (Bisk et al., 2020) is a commonsense QA benchmark for naive physics reasoning, where a solution to a goal must be selected from two choices. Of the training set with 16,113 examples, we use 16,013 for train and 100 for dev. We use the TFDS validation set of 1,838 examples as our test set for reporting numbers.

---

**INPUT**
Caroline never drinks carbonated beverages. Her friends pick on her because of it. One day they challenged her to drink a soda. Caroline wanted to win the challenge.

Predict the next sentence.
OPTIONS:
- Caroline refused to open the soda.
- Caroline opened the soda and drank it all in one gulp!

**TARGET**
Caroline opened the soda and drank it all in one gulp!

---

Table 15: Example input and target for The Story Cloze Test (StoryCloze). StoryCloze (Mostafazadeh et al., 2016) is a commonsense reasoning framework for story generation, where a system chooses the correct ending to a four-sentence story. We use the 2016 version on TFDS. Of the validation set with 1,871 examples (no training set is available), we use 1,671 for train and 200 for dev. We use the TFDS test set of 1,871 examples as our test set for reporting numbers.

## G.4 CLOSED-BOOK QA

| | |
|---|---|
| **INPUT** | |
| What season is the Northern Hemisphere experiencing when it is tilted directly toward the Sun?

OPTIONS:
- fall
- winter
- spring
- summer | |
| **TARGET** | |
| summer | |

Table 16: Example input and target for The AI2 Reasoning Challenge (ARC). ARC (Clark et al., 2018) asks grade-school level 4-way multiple choice science questions. There is a challenge set and an easy set, where the challenge set questions were answered incorrectly by both a retrieval-based algorithm and a co-occurrence algorithm. Of the training sets with 1,119 examples (challenge) and 2,251 (easy), we use we use 919 and 2,051 respectively for train and 200 each for dev. We use the TFDS test sets of 1,172 and 2,376 examples respectively as our test set for reporting numbers.

| |
|---|
| **INPUT** |
| Question: who is the girl in more than you know??
Answer: |
| **TARGET** |
| Romi Van Renterghem. |

Table 17: Example input and target for Natural Questions (Open) (NQ). NQ (Lee et al., 2019; Kwiatkowski et al., 2019) asks for an open-ended answer given a question, where all questions can be answered using the contents of Wikipedia. Of the training set of 87,925 examples, we use 30,000 for train and 200 for dev. We use the TFDS validation set of 3,610 examples as our test set for reporting numbers.

| |
|---|
| **INPUT** |
| Please answer this question: Henry Croft, an orphan street sweeper who collected money for charity, is associated with what organised charitable tradition of working class culture in London, England? |
| **TARGET** |
| pearly kings and queens |

Table 18: Example input and target for Trivia Question Answering (TriviaQA). TriviaQA Joshi et al. (2017) includes question-answer pairs authored by trivia enthusiasts. Of the training set of 87,622 examples, we use 30,000 for train and 200 for dev. We use 7,993 examples from Wikipedia of the 11,313 examples in the TFDS validation set, which is the same validation set used in (Brown et al., 2020). as our test set for reporting numbers.

## G.5 COREFERENCE RESOLUTION

| **INPUT** |
| --- |
| How does the sentence end? |
| |
| Elena wanted to move out of her parents fast but Victoria wanted to stay for a while, |
| |
| OPTIONS: |
| - Elena went to school. |
| - Victoria went to school. |
| **TARGET** |
| Victoria went to school. |

Table 19: Example input and target for Adversarial Winograd Schema Challenge (Winogrande). Winogrande (Sakaguchi et al., 2020) tests for coreference resolution by asking a model to fill in a masked token in a sentence by choosing an entity from two options. Of the 40.4k examples in the XL training set, we use 30,000 for train and 200 for dev. We use the TFDS validation set of 1,267 as our test set for reporting numbers.

| **INPUT** |
| --- |
| Jane knocked on Susan's door, but there was no answer. |
| OPTIONS: |
| - Jane was out. |
| - Susan was out. |
| **TARGET** |
| Susan was out. |

Table 20: Example input and target for Winograd Schema Challenge (WSC273). WSC273 (Levesque et al., 2012) tests for coreference resolution by asking a model to complete the sentence in a fashion that requires understanding the entities in the sentence. Of the 0 examples in the training set (WSC273 is test-set only), we use none for train and none for dev. We use the TFDS test set as our test set for reporting numbers.

## G.6 Reading Comprehension with Commonsense

---

**INPUT**
Complete the passage.

(CNN) – At first glance, "The Flat" might seem like an episode of "Hoarders," Israeli-style. The documentary film opens after an elderly woman dies in Tel Aviv. Her grandchildren assemble to clean out her apartment, packed with dusty books, vintage clothing (dozens of pairs of fancy gloves, for instance), enough purses to stock a department store, jewelry, mementoes and closets full of knickknacks. But buried among the detritus they chance upon something remarkable – mysterious papers linking the grandparents to an important Nazi figure. How could such ardent Zionists, who left their native Germany in the early 1930s, have been involved with an SS official like Leopold von Mildenstein?

What I found out was this journey, the Nazi (

OPTIONS:
- Arnon Goldfinger) and his wife were accompanied by my grandparents," Goldfinger told CNN.
- CNN) and his wife were accompanied by my grandparents," Goldfinger told CNN.
- Germany) and his wife were accompanied by my grandparents," Goldfinger told CNN.
- Israeli) and his wife were accompanied by my grandparents," Goldfinger told CNN.
- Leopold von Mildenstein) and his wife were accompanied by my grandparents," Goldfinger told CNN.
- Nazi) and his wife were accompanied by my grandparents," Goldfinger told CNN.
- SS) and his wife were accompanied by my grandparents," Goldfinger told CNN.
- Tel Aviv) and his wife were accompanied by my grandparents," Goldfinger told CNN.
- The Flat) and his wife were accompanied by my grandparents," Goldfinger told CNN.
- Zionists) and his wife were accompanied by my grandparents," Goldfinger told CNN.

**TARGET**
Leopold von Mildenstein) and his wife were accompanied by my grandparents," Goldfinger told CNN.

---

Table 21: Example input and target for Reading Comprehension with Commonsense Reasoning (ReCoRD). ReCoRD (Zhang et al., 2018) asks for the answer to a cloze-style question where an entity is masked out. Of the the training set of 100,730 examples, we use 30,000 for train and 200 for dev. We use the TFDS validation set of 10,000 examples as our test set for reporting numbers.

## G.7 Translation (7 languages)

---

**INPUT**
Here the largest town of the district is located: Nordenham , lying opposite to Bremerhaven at the Weser mouth.

Translate to German

**TARGET**
An der B 211 befindet sich in Loyermoor der so genannte "Geest-Abbruch", der eine Höhendifferenz von gut 30 Meter überbrückt.

---

Table 22: Example input and output for translation. This example is from WMT'16 English–German; all languages use the same translation templates.

