# OpenReview forum: "Finetuned Language Models are Zero-Shot Learners"
_ICLR.cc/2022/Conference — ICLR 2022 Oral_

### Official Review · Reviewer_BGTp · 2021-10-26

**Correctness:** 3
**Technical Novelty And Significance:** 1
**Empirical Novelty And Significance:** 4
**Recommendation:** 8
**Confidence:** 5

**Main Review:**

While the method is a simple and straightforward scaling up of concepts and ideas from prior works (e.g. Zhong et al, Adapting ...; Mishra et al, cross-task generalization ...), the empirical results are thorough and impressive (outperforming GPT-3 with a slightly smaller model). The analyses also helps us understand when this method would work and inform us about future research directions.

Below are my concrete questions and comments:

**Additional Tasks Results (3.4)**

In Appendix A.1, the paper mainly draws conclusions based on comparisons between GPT-3 and FLAN, which I do not think are fair: GPT-3 and FLAN differ in model size and pre-training data distribution. Instead, I think Base LM vs. FLAN might be a better comparison between “Off the shelf LM” and “instruction-tuned” model (though it won’t change the conclusion).

It is also worth pointing out in the main paper that for most of the additional tasks, even though it does not lead to higher accuracy, the performance of FLAN is still at least comparable (e.g. <1% worse and difference generally negligible) to Base-LM 0-shot. The only outlier seems to be ReCorD where the performance drops significantly after instruction-tuning, and this probably deserves some discussion.

Also I might have missed it - for the Base LM 137B zero-shot result, is it on the average template, or the best template?

**Number of Task Clusters (section 4.1)**

For Figure 5 can you add the untuned model to the curve with the x-axis=0 (0-task cluster)? This can help us understand how much even 1 cluster (e.g. summarization) may help.

**Explanation for scaling (section 4.2)**

It is an insightful empirical result that instruction tuning only works when model size reaches 68B. However, I am not entirely sure about the potential explanation of “model capacity”. There might be two potential explanations to this phenomena: 1) “model capacity”: as the paper has mentioned, smaller pre-trained models do not have enough model capacity and underfit the instruction tuning data, and 2) “better OOD generalization”: better quality pre-trained models have higher OOD generalization ability (and OOD accuracy) and they are less likely to “overfit” to in-distribution data.

I personally find the second explanation more convincing. For example, Sahn et al. (https://arxiv.org/abs/2110.08207) finds that even models with only 11B parameters can generalize to unseen tasks, using T5 (MLM) and a larger set of prompts. The use of MLM objectives (might) improve the pre-training quality, while more prompts reduce the “overfitting to in-domain data” issue.

I appreciate the fact that the author explicitly states the model capacity hypothesis more as a conjecture rather than a solid explanation. It’d be great if the authors can support the explanation further with more empirical evidence. On the other hand, however, since the results from Sanh et al came out only 2 weeks ago,  I would not change the score based on the response to this question.

**In-context Few-shot vs. Fine-tuned Few-shot (Section 4.3)**

Can the authors compare “fine-tuning/prefix-tuning an instruction tuned model with 16 examples” (appendix C but with only 16 examples) with “in-context prompting” (in 4.3 of the main paper), similar to Chen et al. (https://arxiv.org/abs/2110.07814 )? This would further inform us how we should use the few-shot learning examples for larger language models: put it in-context, or fine-tune? Again, since the comparison of Chen et al. came out only 2 weeks ago and the paper limit is 9 pages, I would not change the score based on the response to this question.

**Others**

Results of Appendix C are interesting and potentially impactful - this might imply that instruction-tuned models will become the new “base” model for the pretraining-finetuning paradigm. Is it possible to briefly mention it in the main paper as well (and redirect the readers to the appendix to see the full results)?

It might be too late to change the name, but “Finetuned LAnguage Net” (FLAN) is uninformative, since it does not capture any unique aspect of this method. What does “LAnguage” mean here, "natural language instruction" or "language model"? If it is the former, then directly including the word “instruction” might be better; and hopefully it’s not the latter, since even fine-tuned BERT on SST-2 counts as a fine-tuned language model ...

**Typo**
Intro:
Instruction tuning is “a” simple method that, as ….

Conclusion:
Moreover, our work supercedes recent work such “as”


**Summary Of The Paper:**

The paper proposes a simple method, "instruction-tuning", to improve the zero-shot learning capability of large language model, which 1) annotates prompts for a wide range of tasks and then 2) fine-tunes the model to "answer/respond to" those prompt. The empirical results are impressive: after instruction-tuning, the 0-shot performance is better than GPT-3 (0-shot, sometimes few-shot) on a wide range of datasets; nevertheless, on datasets with formats already similar to language modeling, the performance gain is negligible or even negative.

The paper also made a few other observations 1) performance benefits from the number of task clusters 2) instruction-tuning is only beneficial when the model size is larger enough, and 3) few-shot learning still helps.

**Summary Of The Review:**

While the method is not new, the empirical results are strong and comprehensive. Though I disagree on the interpretation of some empirical results, overall the additional analyses bring us further insights on what method works for very large language models (i.e. > 100B dense model). I highly recommend the paper to be accepted to ICLR 2022.

---

> ### Author Response · Authors · 2021-11-23
> **Reply to Reviewer BGTp**
>
> Thank you for the detailed review and the positive feedback! We aimed to propose a simple and intuitive method for improving the zero-shot performance of language models, and we are glad you found the empirical results thorough and impressive. We revised the paper below based on your comments—please let us know if you have any further clarifications or suggestions to improve the paper!
>
> > In Appendix A.1, Base LM vs FLAN might be a better comparison.
>
> Indeed, Base LM vs FLAN is the most relevant comparison for instruction tuning. We originally included the comparison to GPT-3 because readers were more likely to be familiar with the performance of GPT-3, but in the revision we added more text to highlight the Base LM vs FLAN comparison. Thank you for suggesting this and pointing out that the conclusions do not change much.
>
> > Which prompts do the Base LM results use?
>
> Base LM is similar to GPT-3 in that it is a pretrained language model without any instruction tuning. Hence, we use the same prompts as GPT-3 to evaluate Base LM. The GPT-3 paper only provided one prompt per dataset. We added this explanation to the first question/answer of the new FAQ section in the Appendix.
>
> > For Figure 5 can you add the untuned model to the curve with the x-axis=0 (0-task cluster)?
>
> The reason we did not plot the untuned model (Base LM) on the curve at x=0 is that the Base LM model uses different prompts than instruction-tuned models (Base LM uses GPT-3 style language modeling prompts, whereas FLAN responds to instructions) and thus it would not be an apples-to-apples comparison. However, we added them in as horizontal lines labeled with “Base LM”.
>
> > Discussion in light of Sahn et al.
>
> Thank you for pointing out this recent related paper (and for appropriately noting that it came out two weeks ago). We added this to our extended related work section (E.6) (as well as another recent related work, “MetaICL: Learning to Learn In Context”, which was posted on arxiv less than a month ago). Indeed, our model capacity explanation is just a conjecture, and it is challenging to compare the results across these papers because of differences such as model sizes, model types (decoder-only vs encoder-decoder), pretraining data, task mixtures, and type of instructions.
>
> > Results of Appendix C are interesting… this might imply that instruction-tuned models will become the new “base” model… is it possible to briefly mention it in the main paper?
>
> Thank you for this helpful suggestion! We added a pointer to it in the first paragraph of the Discussion section.
>
> > Can the authors compare “fine-tuning/prefix-tuning an instruction tuned model with 16 examples” (appendix C but with only 16 examples)?
>
> As the focus of our paper is zero-shot learning, we prioritized our computational resources during the response period for the ablation study requested by Reviewers 4nJJ and K8bM. However, we appreciate the suggestion and you appropriately pointing out that Chen et al. came out after the ICLR submission deadline—we added this to the paper as further work.

---

> > ### Comment · Reviewer_BGTp · 2021-11-29
> > **Thanks for the reply!**
> >
> > Thanks for the reply! I have read the reply and other reviewers' comment and my decision is unchanged.

---

### Official Review · Reviewer_K8bM · 2021-10-31

**Correctness:** 4
**Technical Novelty And Significance:** 2
**Empirical Novelty And Significance:** 4
**Recommendation:** 8
**Confidence:** 3

**Main Review:**

----
Detailed comments:
----

- "For each dataset, we manually compose ten unique templates":  Why not have templates per task cluster instead of per dataset?  it is likely a relatively minor effect given the results from Appendix B but it seems like it could slightly prevent overfitting

- The ablation in 4.1 was great (number of clusters).  Nit: I would have tried to move the (datasets per cluster/templates per dataset) ablation to the main body as well and shortened Section 3

- The 4.2 (scaling laws) ablation is perhaps the most interesting of all.

- In figure 6A, why was performance not increasing for untuned models w.r.t model size?  This seems to contradict findings from Brown et al where larger models did better on essentially all tasks.  Were there perhaps some poor datasets that happened to be in the held-in split (since the held-out tasks don't seem to have the same trend)?

----
Appendix:
----

I liked the section B ablations (as implied above).

That more templates per dataset didn't help is particularly interesting and suggests some questions. You hypothesize that more templates doesn't help because "models at such scale do not easily overfit to a finetuning single task" - but my intuition is for an opposite explanation -- that the models at such scale easily memorize a small number of templates!  One may even wonder if the instruction nature of the templates is helping at all.

From what I can tell, Appendix C on prompt tuning (which is very interesting) is maybe the primary evidence the instructions are important.  I think more could be done here, some ideas, probably there are better ways to test:
- Have templates that leave out "instructions":  I would guess it wouldn't affect held-in task performance much, but would affect held-out tasks.
- Consider HellaSwag/PiQA/etc, where FLAN underperformed few-shot and even zero-shot.  One might hypothesize that if using a (subotimal) template that is less natural for language modeling, that zero-shot performance would suffer, but that FLAN performance wouldn't
- One might hypothesize that the "turn the task around" templates help more than the other more straightforward templates that don't swap information between the prompt and response.
- Easy but probably not great thing to try:  held-out tasks with wrong/useless templates

A final thought:  It's not obvious that using as many training examples per dataset as possible is optimal, given that the model could overfit to dataset-specific spurious correlations.  This could be another area to investigate

----
Misc:
----

- UnifiedQA seems potentially worth citing as prior work

**Summary Of The Paper:**

The paper creates a dataset of over 60 NLP tasks described via instructions (using templates for each task) and finds this boosts zero-shot performance on unseen tasks.

**Summary Of The Review:**

Overall, the paper's idea is powerful (but of somewhat limited novelty) and the results are good (but not great).  Its greatest strength IMO was the ablations.  My biggest complaint is that it's not completely clear the instructions themselves are important at all - I suggest a few more experiments, though they don't seem crucial.

---

> ### Author Response · Authors · 2021-11-23
> **Reply to Reviewer K8bM**
>
> Thank you for the insightful review. Our paper proposed an approach that substantially improves zero-shot performance of language models, and we are glad that you liked the ablations and the prompt tuning experiments!
>
> > My biggest complaint is that it's not completely clear the instructions themselves are important at all…have templates that leave out "instructions"
>
> *(This response is identical to the response given to Reviewer 4nJJ.)*
>
> Thank you for this thoughtful comment! We performed an additional experiment for this that shows that including instructions during finetuning does indeed allow the model to better perform zero-shot tasks.
> We added it to Appendix B2 in the revised paper and also summarize it below for convenience.
>
> To tease apart the role of instructions, we considered several setups where the instructional prompts were removed during training.
>
> 1. **No template**: Only the inputs and outputs were given to the model, as would be the case for a task-specific model. For example, for translation to French, the input would be *“the dog runs”* and the output would be *“le chien court”*. This does not distinguish among tasks during multi-task training.
> 2. **Task/dataset name**: In this setup, each input is prepended with the name of the task and the name of the dataset. For example, for translation to French, the input would be *“[Translation: WMT’14 to French] The dog runs”*.
>
> We compare these two setups to FLAN’s finetuning procedure, which used natural instructions (e.g., *“‘The dog runs.’ Translate this sentence to French.”*).
>
> For no template finetuning, during zero-shot inference we use the natural instructions from FLAN (otherwise, the model would not know what task to perform). For task/dataset name finetuning, during zero-shot inference we used both the natural instructions from FLAN as well as task/dataset name. The results are shown in the table below:
>
> | Finetuning prompt | Inference prompt | Read. Comp. | Closed-B QA | NLI | Translation | 4-task Avg.
> | ----------- | ----------- | ----------- |  ----------- |  ----------- |  ----------- |  ----------- |
> | Natural instructions (=FLAN) | Natural instructions | 77.4 | 56.6 | 56.2 | 30.7 | **55.2** |
> | No template | Natural instructions | 58.2 | 25.5 | 50.2 | 15.0 | **37.3** |
> | Task/dataset name | Task/dataset name | 64.9 | 40.8 | 60.2 | 21.9 | **47.0** |
> | Task/dataset name | Natural instructions | 63.0 | 44.8 | 52.8  | 25.9 | **46.6** |
>
> (Training with task/dataset name achieved a high NLI score largely because it achieved a score of 83.9 on the CB dataset, for which the validation set only has 56 examples. FLAN also gets 83.9 with the best dev template, but the average template was only 64.1.)
>
> Finetuning with no templates has very poor performance, which is expected since the model is not able to distinguish between tasks during finetuning. Finetuning with the task/dataset name improves performance noticeably over no template (using task/dataset name for inference versus natural instructions for inference performed about the same in aggregate), but is still almost eight points behind FLAN on average across the four task clusters. This result indicates that finetuning with instructions is crucial for zero-shot performance on unseen tasks.

---

> > ### Author Response · Authors · 2021-11-23
> > **Clarifications**
> >
> > > Why not have templates per task cluster instead of per dataset?
> >
> > We created unique templates per dataset because datasets in the same task cluster often differed slightly (e.g., “is this movie review positive” vs “is this yelp review positive”). The complete list of templates per dataset is given in the supplementary material. We added this clarification to the FAQ section in the appendix of the paper for additional visibility.
> >
> > > In Figure 6A, why was performance not increasing for untuned models w.r.t model size?
> >
> > *(This response is identical to the response given to Reviewer 4nJJ.)*
> >
> > For context, Figure 6A is a check of correctness for Figure 6B. Figure 6A confirms that scale improves performance for tasks that were seen during instruction tuning, as expected. The untuned Base LM model performance in 6A is shown just for completeness.
> >
> > Nonetheless, the fact that scale does not always improve the zero-shot performance of untuned Base LM is an interesting artifact. Initially, we were surprised, because Brown et al., 2020 shows that scale improves performance across a large number of tasks in aggregate.
> >
> > It turns out that scale does not improve performance for certain tasks. This is especially true for zero-shot learning, and we think that this happens to be the case for the reading comprehension and sentiment analysis tasks we evaluate. The GPT-3 paper itself similarly reports that zero-shot performance on BoolQ and DROP decreases from 13B to 175B parameters. The GPT-3 paper does not show results on sentiment analysis, but Holtzman et al., 2021 find that zero-shot performance on SST-2 also gets worse from 13B to 175B parameters. Hence, this artifact is consistent across both GPT-3 and the models we use.
> >
> > This artifact is certainly worth further study, but is outside the scope of instruction tuning. Ideally, we would have performed the Figure 6 ablation with cross-validation instead of a single split, which likely would have smoothed out that artifact. We added this explanation to the new FAQ section in the paper.
> >
> >
> > > More templates didn’t help is particularly interesting… what about the opposite explanation that large models easily memorize a small number of templates?
> >
> > This hypothesis was our original motivation for writing ten templates per dataset, and we were surprised to see in the Appendix B ablations that this did not make a substantial difference in performance when there was a large number of tasks. We think this result certainly warrants further investigation, but our intitution is that larger pretrained models, which are more data efficient, are better able to generalize.
> >
> > > UnifiedQA seems potentially worth citing.
> >
> > Thanks for pointing us to this! We added it to the related work section.

---

### Official Review · Reviewer_DjKD · 2021-11-02

**Correctness:** 4
**Technical Novelty And Significance:** 3
**Empirical Novelty And Significance:** 3
**Recommendation:** 8
**Confidence:** 4

**Main Review:**

Pros:
1. The problem addressed has high practical value: it tries to make large pre-trained language model more accessible to a range of NLP tasks. The "instruction tuning" idea will significantly reduce the cost for task-specific fine tuning, labeled data and prompt engineering compared to other approaches.
2. The method is simple and easy to implement. Authors carefully design the experiment to minimize the leakage between the fine-tuning and inference data. Given that, it still shows superior performance on different types of NLP tasks. The result on specific task can be further improved when adapting with "prompt tuning" on labeled data, which shows that the instruction-tuning process does not drop much task-specific knowledge from the original pretrained model.
3. The analysis presented in the main paper and the appendix is thorough enough. Authors also discussed about the limitation of model when downstream tasks are more similar to language modeling tasks.

Cons:
There are still a few questions that can be addressed to make the analysis comprehensive.
1. Have authors try to use the FLAN prompts on GPT3 or BaseLM and how does the performance look like?
2. Since instruction tuning will adjust all the parameters in the original pre-trained language model, there is a question what about what is the potential impact of this tuning process? Will it drops any knowledge of any tasks, which will be a disadvantage when the task's labeled data is available? In the Analysis C in the appendix, it will be good to have results for tasks other than classification such as summarization or question answering; and also to have a baseline where the BaseLM model is fine-tuned directly with the task labeled data (without prompt/soft-prompt).


**Summary Of The Paper:**

The paper explores a simple and effective method to improve zero-shot performance of pretrained language models. Authors take a 137B parameter pretrained model and finetune it on multiple tasks verbalized via natural language instruction templates. As the result, the instruction-tuned model performs well on un-seen tasks with the zero-shot setting.

**Summary Of The Review:**

Overall, the paper proposed an interesting idea and showed strong empirical results, hence I vote for accepting.

---

> ### Author Response · Authors · 2021-11-23
> **Reply to Reviewer DjKD**
>
> Thank you for the detailed review and positive feedback! We proposed a method to improve the zero-shot performance of language models, and we are glad you found instruction tuning to be simple, effective, and of high practical value. We discuss your comments below. Please let us know if you have any additional suggestions to improve the paper!
>
> > Do FLAN prompts work for GPT3 or Base LM?
>
> A small but noteworthy difference between FLAN prompts and GPT-style prompts is that FLAN prompts are formulated as responding to an instruction, whereas GPT-style prompts are formatted as continuations of sentences. For instance, a FLAN prompt might be *“The dog runs. Translate this sentence to French.”,* whereas a GPT-style prompt might be *“‘The dog runs’ translated to French is:”.*
> For this reason, FLAN prompts do not work well for pretrained language models without finetuning, so we did not report these results in the paper. Performance was near zero for most generation tasks. For instance, given the FLAN prompt *“The dog runs. Translate this sentence to French.”,* Base LM continues with *“The dog runs after the cat”* instead of actually translating the sentence. Hence, we used the established GPT-3 prompts for our Base LM baselines. For additional visibility, we added this to the new FAQ section in the Appendix.
>
> > Since instruction tuning will adjust all the parameters in the original pre-trained language model, there is a question what about what is the potential impact of this tuning process?
>
> The goal of instruction tuning is to allow models to follow instructions for a range of tasks, including both seen and unseen tasks, and the way we do this modifies all parameters in the language model. Intuitively, we might expect instruction tuning to lead to worse perplexity on pure language modeling, as presumably the model is optimized for responding to NLP tasks instead of the original pretraining objective. We did not measure FLAN’s perplexity on the original pretraining data compared with Base LM, but for language modeling tasks shown in Appendix A.1, FLAN performs about the same or slightly worse.
>
> > Will [FLAN] drop any knowledge of any tasks, which will be a disadvantage when the task's labeled data is available?
>
> Interpreting this question as whether a massively multi-task model performs as well as a model finetuned on only a single task, we think this is still an open question. This likely depends on the specific mixture of tasks used in instruction tuning, since many similar tasks will likely lead to positive task transfer. An experiment could compare our FLAN model with a single-task model on a seen task, but we will leave this for future work since our paper focuses on zero-shot learning.
>
> > In the Analysis C in the appendix, it will be good to have results for tasks other than classification such as summarization or question answering, and also to have a baseline where the Base LM model is fine-tuned directly with the task labeled data (without prompt/soft-prompt).
>
> We decided to prioritize our compute during the response period for the ablation study requested by two other reviewers (as prompt tuning / supervised learning is not the focus of this paper). However, thank you for the suggestion—we added this suggestion into the the Analysis C in the appendix as future work!

---

### Official Review · Reviewer_4nJJ · 2021-11-06

**Correctness:** 3
**Technical Novelty And Significance:** 4
**Empirical Novelty And Significance:** 4
**Recommendation:** 8
**Confidence:** 4

**Main Review:**

Overall well-written with compelling results, this paper describes a new language model (FLAN) and shows how it improves upon the zero-shot task performance of previous language models such as GPT-3. While the paper is lacking some additional analysis, I am hesitant to recommend extremely compute-intensive ablations due the large size of the model (137B parameters).

Strengths:
 - Considers a reasonably wide set of 62 datasets; although the inherent arbitrariness in dataset clustering was listed as a limitation, the clusters look quite reasonable to me, and the removal of overlapping datasets (e.g., "Reading Comprehension w/ Commonsense") seems appropriate.
 - Results are better than a strong Base LM baseline, as well as existing state-of-the-art models (GPT-3)
 - Overall the approach is intuitive and conceptually compelling
 - Highly relevant to ongoing work on language modeling, prompt tuning, and zero-shot learning

Weaknesses:
 - From these experiments, it is unclear whether models are actually "learning to follow instructions" or just learning a very large space of tasks from the fine-tuning procedure. In other words, even though prompt variance is reported at inference time, the models could potentially perform just as well with nonsense or missing prompts during fine-tuning. As far as I can tell, no experiments that rule out this possibility exist.
 - Although qualitatively useful, the analysis in 4.1 does not conclusively show that the number of instruction tuning clusters aids performance, or that this trend is likely to continue with more clusters. Most of the gain could be acquired by tasks which are most difficult, or most similar to the heldout task, and this analysis cannot disprove such an interpretation. A proper analysis would consider more heldout tasks and permutations of training data, but presumably this is prohibitively expensive.
 - The paper is missing important details about hardware usage and training time
 - Some possible issues which might be resolved by the additional questions below

Additional Questions:
 - "For each dataset, we manually compose ten unique templates that use natural language instructions to describe the task for that dataset." Do you have unique prompts for each dataset or only for each dataset cluster? Based on a cursory look at the supplementary material, I would assume the latter.
 - I didn't fully understand the justification for the OPTIONS token. Are the fine-tuned models successfully putting (almost) all of their probability mass on the corresponding options? How is the Base LM evaluated (if it's not fine-tuned, presumably it doesn't learn how to handle these options)?
 - Figure 6A: why does the untuned model see worse performance with more parameters?

Nits:
 - Figure 1 (Bottom) is possibly misleading, since AFAICT zero-shot FLAN underperforms few-shot GPT-3 on the majority of tasks
 - Not clear what "turning the task around" means for some tasks, or why this is a useful type of prompt diversity

**Summary Of The Paper:**

This paper describes an approach to fine-tuning large language models which can improve zero-shot accuracy on unseen tasks.

**Summary Of The Review:**

I give this paper a strong recommendation, in spite of some missing ablations.

---

> ### Author Response · Authors · 2021-11-22
> **Reply to Reviewer 4nJJ**
>
> Thank you for the insightful review! Our paper explores an instruction-tuned model that outperforms BaseLM and GPT-3, and we are glad you found our approach intuitive and conceptually compelling. Below is discussion on the points you brought up. Please let us know if you have any further suggestions on how to improve the paper!
>
> >  It is unclear whether models are actually "learning to follow instructions" or just learning a very large space of tasks from the fine-tuning procedure…models could potentially perform just as well with nonsense or missing prompts during fine-tuning.
>
> *(This response is identical to the response given to Reviewer K8bM.)*
>
> Thank you for this insightful comment! We performed an additional experiment for this that shows that including instructions during finetuning does indeed allow the model to better perform zero-shot tasks.
> We added it to Appendix B2 in the revised paper and also summarize it below for convenience.
>
> To tease apart the role of instructions, we considered several setups where the instructional prompts were removed during training.
>
> 1. **No template**: Only the inputs and outputs were given to the model, as would be the case for a task-specific model. For example, for translation to French, the input would be *“the dog runs”* and the output would be *“le chien court”*. This does not distinguish among tasks during multi-task training.
> 2. **Task/dataset name**: In this setup, each input is prepended with the name of the task and the name of the dataset. For example, for translation to French, the input would be *“[Translation: WMT’14 to French] The dog runs”*.
>
> We compare these two setups to FLAN’s finetuning procedure, which used natural instructions (e.g., *“‘The dog runs.’ Translate this sentence to French.”*).
>
> For no template finetuning, during zero-shot inference we use the natural instructions from FLAN (otherwise, the model would not know what task to perform). For task/dataset name finetuning, during zero-shot inference we used both the natural instructions from FLAN as well as task/dataset name. The results are shown in the table below:
>
> | Finetuning prompt | Inference prompt | Read. Comp. | Closed-B QA | NLI | Translation | 4-task Avg.
> | ----------- | ----------- | ----------- |  ----------- |  ----------- |  ----------- |  ----------- |
> | Natural instructions (=FLAN) | Natural instructions | 77.4 | 56.6 | 56.2 | 30.7 | **55.2** |
> | No template | Natural instructions | 58.2 | 25.5 | 50.2 | 15.0 | **37.3** |
> | Task/dataset name | Task/dataset name | 64.9 | 40.8 | 60.2 | 21.9 | **47.0** |
> | Task/dataset name | Natural instructions | 63.0 | 44.8 | 52.8  | 25.9 | **46.6** |
>
> (Training with task/dataset name achieved a high NLI score largely because it achieved a score of 83.9 on the CB dataset, for which the validation set only has 56 examples. FLAN also gets 83.9 with the best dev template, but the average template was only 64.1.)
>
> Finetuning with no templates has very poor performance, which is expected since the model is not able to distinguish between tasks during finetuning. Finetuning with the task/dataset name improves performance noticeably over no template (using task/dataset name for inference versus natural instructions for inference performed about the same in aggregate), but is still almost eight points behind FLAN on average across the four task clusters. This result indicates that finetuning with instructions is crucial for zero-shot performance on unseen tasks.

---

> > ### Author Response · Authors · 2021-11-22
> > **Clarifications**
> >
> > > Missing details about hardware usage and training time.
> >
> > Instruction-tuning our 137B parameter model for 30k steps took around 60 hours on a TPUv3 with 128 cores. Thank you for pointing this out—we added this detail into the manuscript.
> >
> > > Do you have unique prompts for each dataset or only for each [task] cluster?
> >
> > The ten unique templates are for each dataset and not for a task cluster. This is because datasets in the same task cluster often differed slightly (e.g., *”is this movie review positive”* vs *“is this yelp review positive”*). The complete list of templates per cluster is given in the supplementary material. We added this clarification to the FAQ section in the appendix of the paper for additional visibility.
> >
> > > Further details on use of OPTIONS?
> >
> > The motivation for the OPTIONS token is to allow the language model to learn to distribute probability mass over a known set of outputs. Borrowing the example from Holtzman et al., 2021, in vanilla rank classification, given an input *“People left the party because…”*, the probability mass on the *“it was 2am”* continuation was diluted by similar continuations such as *“it was 1am”*, *“it was 3am”*, etc. Hence, telling the model beforehand about the classification options tells it which continuations to give probability mass to. Base LM is not evaluated using OPTIONS, just rank classification. We did not perform a quantitative evaluation of adding up the probabilities assigned to all options, but for most tasks, the top output via greedy sampling when using OPTIONS was one of the provided options.
> >
> > > Figure 6A: why does the untuned model see worse performance with more parameters?
> >
> > (This response is identical to the response given to Reviewer K8bM.)
> >
> > For context, Figure 6A is a check of correctness for Figure 6B. Figure 6A confirms that scale improves performance for tasks that were seen during instruction tuning, as expected. The untuned Base LM model performance in 6A is shown just for completeness.
> >
> > Nonetheless, the fact that scale does not always improve the zero-shot performance of untuned Base LM is an interesting artifact. Initially, we were surprised, because Brown et al., 2020 shows that scale improves performance across a large number of tasks in aggregate.
> >
> > It turns out that scale does not improve performance for certain tasks. This is especially true for zero-shot learning, and we think that this happens to be the case for the reading comprehension and sentiment analysis tasks we evaluate. The GPT-3 paper itself similarly reports that zero-shot performance on BoolQ and DROP decreases from 13B to 175B parameters. The GPT-3 paper does not show results on sentiment analysis, but Holtzman et al., 2021 find that zero-shot performance on SST-2 also gets worse from 13B to 175B parameters. Hence, this artifact is consistent across both GPT-3 and the models we use.
> >
> > This artifact is certainly worth further study, but is outside the scope of instruction tuning. Ideally, we would have performed the Figure 6 ablation with cross-validation instead of a single split, which likely would have smoothed out that artifact. We added this explanation to the new FAQ section in the paper.
> >
> > > Figure 1 (Bottom) is possibly misleading, since AFAICT zero-shot FLAN underperforms few-shot GPT-3 on the majority of tasks.
> >
> > Thank you for pointing this out. Although Figure 1 bottom is technically correct for the three task clusters shown, zero-shot FLAN does not outperform few-shot GPT-3 on task clusters that are not shown in the figure (e.g., translation, commonsense reasoning). Hence, to reduce the risk of misleading the reader that this result extrapolates to tasks not shown, we changed the caption to “Performance of zero-shot FLAN, compared with zero-shot and few-shot GPT-3, on three unseen task types where instruction tuning improved performance substantially out of ten we evaluate.”
> >
> > > Not clear what "turning the task around" means for some tasks, or why this is a useful type of prompt diversity.
> >
> > Regarding templates that “turn the task around”, the motivation was that it would increase the number of tasks that the model could perform. We did not do this for all tasks (all templates are shown in the Supplementary Material). While outside of the scope of our evaluation, these templates could potentially be useful for open-ended generation tasks. We also added some examples of FLAN’s responses to open-ended generation tasks in Appendix G.

---

> > > ### Comment · Reviewer_K8bM · 2021-11-29
> > > **Thanks for the reply**
> > >
> > > The new appendix B2 results are great and address my primary concern.  One final curiosity - did the fine-tuning variants do just as well on held-in datasets?
> > >
> > > Overall I still mostly stand by my original assessment of the paper.  I think the paper is still a good paper, and probably would be in the top 10-20% of accepted papers according to my tastes.
> > >
> > > (Edit: oops, replied to the wrong place...)

---

### Author Response · Authors · 2021-11-26
**Response to reviews (TLDR)**

We thank all four reviewers for the comprehensive feedback. As a summary, our work is about the striking ability for finetuned language models at scale to follow instructions. We present empirical results across 10 NLP tasks, with ablation studies that tease apart the key components of instruction tuning. Positive cross-task generalization results imply that instructional finetuning at scale can play a key role in developing generalist models.

All four reviews were positive. Two reviewers, however, brought up the insightful observation about whether instructions are really needed: could the model achieve the same zero-shot performance via multitask finetuning without instructions? To answer this question, we performed an ablation study by finetuning two such models that did not have instructions. One model was finetuned without any templates indicating the task, and another model was finetuned where each example had the task/dataset name prepended to the input. We then evaluated these models using the same task-split setup as the paper, and found that they performed well below the original model on average (~8 points lower on average across 19 datasets), demonstrating that instructions do indeed matter during finetuning. We added this ablation study into Appendix B.2 and also explain it in the individual responses to reviewers 1 and 3. For the additional points brought up by reviewers, we replied in the individual responses to reviewers and also revised the paper accordingly.

(We meant to post this tldr during the response period. Sorry for the additional ping!)

---

### Decision · Program_Chairs · 2022-01-20

**Decision:**

Accept (Oral)

**Comment:**

This paper examines the extent to which a large language model (LM) can generalize to unseen tasks via "instruction tuning", a process that fine-tunes the LM on a large number of tasks with natural language instructions.  At test time, the model is evaluated zero-shot on held out tasks.  The empirical results are good, and the 137B FLAN model generally out performs the 175B untuned GPT-3 model.

All reviewers voted to accept with uniformly high scores, despite two commenting on the relative lack of novelty.  The discussion period focused on questions raised by two reviewers regarding the usefulness of fine-tuning with instructions vs. multi-task fine-tuning without instructions.  The authors responded with an ablation study demonstrating that providing instructions at during tuning led to large gains.

Overall the paper's approach and detailed experiments will be useful for other researchers working in this fast moving area in NLP.